# Wasserstein Auto-encoded MDPs
## Formal Verification of Efficiently Distilled RL Policies with Many-sided Guarantees

**Florent Delgrange**
AI Lab, Vrije Universiteit Brussel (VUB)
University of Antwerp
florent.delgrange@ai.vub.ac.be

**Ann Nowé**
AI Lab, VUB

**Guillermo A. Pérez**
University of Antwerp
Flanders Make

## Abstract

Although deep reinforcement learning (DRL) has many success stories, the large-scale deployment of policies learned through these advanced techniques in safety-critical scenarios is hindered by their lack of formal guarantees. Variational Markov Decision Processes (VAE-MDPs) are discrete latent space models that provide a reliable framework for distilling formally verifiable controllers from any RL policy. While the related guarantees address relevant practical aspects such as the satisfaction of performance and safety properties, the VAE approach suffers from several learning flaws (posterior collapse, slow learning speed, poor dynamics estimates), primarily due to the absence of abstraction and representation guarantees to support latent optimization. We introduce the Wasserstein auto-encoded MDP (WAE-MDP), a latent space model that fixes those issues by minimizing a penalized form of the optimal transport between the behaviors of the agent executing the original policy and the distilled policy, for which the formal guarantees apply. Our approach yields bisimulation guarantees while learning the distilled policy, allowing concrete optimization of the abstraction and representation model quality. Our experiments show that, besides distilling policies up to 10 times faster, the latent model quality is indeed better in general. Moreover, we present experiments from a simple time-to-failure verification algorithm on the latent space. The fact that our approach enables such simple verification techniques highlights its applicability.

## 1 Introduction

*Reinforcement learning* (RL) is emerging as a solution of choice to address challenging real-word scenarios such as epidemic mitigation and prevention strategies (Libin et al., 2020), multi-energy management (Ceusters et al., 2021), or effective canal control (Ren et al., 2021). RL enables learning high performance controllers by introducing general nonlinear function approximators (such as neural networks) to scale with high-dimensional and continuous state-action spaces. This introduction, termed *deep-RL*, causes the loss of the conventional convergence guarantees of RL (Tsitsiklis, 1994) as well as those obtained in some continuous settings (Nowe, 1994), and hinders their wide roll-out in critical settings. This work *enables* the *formal verification* of *any* such policies, learned by agents interacting with unknown, continuous environments modeled as *Markov decision processes* (MDPs). Specifically, we learn a *discrete* representation of the state-action space of the MDP, which yield both a (smaller, explicit) *latent space model* and a distilled version of the RL policy, that are tractable for *model checking* (Baier & Katoen, 2008). The latter are supported by *bisimulation guarantees*: intuitively, the agent behaves similarly in the original and latent models. The strength of our approach is not simply that we verify that the RL agent meets a *predefined* set of specifications, but rather provide an abstract model on which the user can reason and check *any* desired agent property.

*Variational MDPs* (VAE-MDPs, Delgrange et al. 2022) offer a valuable framework for doing so. The distillation is provided with PAC-verifiable bisimulation bounds guaranteeing that the agent behaves similarly (i) in the original and latent model (*abstraction quality*); (ii) from all original states embedded to the same discrete state (*representation quality*). Whilst the bounds offer a confidence metric that enables the verification of performance and safety properties, VAE-MDPs suffer from several learning flaws. First, training a VAE-MDP relies on variational proxies to the bisimulation bounds, meaning there is no learning guarantee on the quality of the latent model via its optimization. Second, *variational autoencoders* (VAEs) (Kingma & Welling, 2014; Hoffman et al., 2013) are known

to suffer from *posterior collapse* (e.g., Alemi et al. 2018) resulting in a deterministic mapping to a unique latent state in VAE-MDPs. Most of the training process focuses on handling this phenomenon and setting up the stage for the concrete distillation and abstraction, finally taking place in a second training phase. This requires extra regularizers, setting up annealing schemes and learning phases, and defining prioritized replay buffers to store transitions. Distillation through VAE-MDPs is thus a meticulous task, requiring a large step budget and tuning many hyperparameters.

Building upon *Wasserstein* autoencoders (Tolstikhin et al., 2018) instead of VAEs, we introduce *Wasserstein auto-encoded MDPs* (WAE-MDPs), which overcome those limitations. Our WAE relies on the *optimal transport* (OT) from trace distributions resulting from the execution of the RL policy in the real environment to that reconstructed from the latent model operating under the distilled policy. In contrast to VAEs which rely on variational proxies, we derive a novel objective that directly incorporate the bisimulation bounds. Furthermore, while VAEs learn stochastic mappings to the latent space which need be determinized or even entirely reconstructed from data at the deployment time to obtain the guarantees, our WAE has no such requirements, and learn *all the necessary components to obtain the guarantees during learning*, and does not require such post-processing operations.

Those theoretical claims are reflected in our experiments: policies are distilled up to 10 times faster through WAE- than VAE-MDPs and provide better abstraction quality and performance in general, without the need for setting up annealing schemes and training phases, nor prioritized buffer and extra regularizer. Our distilled policies are able to recover (and sometimes even outperform) the original policy performance, highlighting the representation quality offered by our new framework: the distillation is able to remove some non-robustness of the input RL policy. Finally, we formally verified *time-to-failure* properties (e.g., Pnueli 1977) to emphasize the applicability of our approach.

**Other Related Work.** Complementary works approach safe RL via formal methods (Junges et al., 2016; Alshiekh et al., 2018; Jansen et al., 2020; Simão et al., 2021), aimed at formally ensuring safety *during RL*, all of which require providing an abstract model of the safety aspects of the environment. They also include the work of Alamdari et al. (2020), applying synthesis and model checking on policies distilled from RL, without quality guarantees. Other frameworks share our goal of verifying deep-RL policies (Bacci & Parker, 2020; Carr et al., 2020) but rely on a known environment model, among other assumptions (e.g., deterministic or discrete environment). Finally, *DeepSynth* (Hasanbeig et al., 2021) allows learning a formal model from execution traces, with the different purpose of guiding the agent towards sparse and non-Markovian rewards.

On the latent space training side, WWAEs (Zhang et al., 2019) reuse OT as latent regularizer discrepancy (in Gaussian closed form), whereas we derive two regularizers involving OT. These two are, in contrast, optimized via the dual formulation of Wasserstein, as in *Wassertein-GANs* (Arjovsky et al., 2017). Similarly to *VQ-VAEs* (van den Oord et al., 2017) and *Latent Bernoulli AEs* (Fajtl et al., 2020), our latent space model learns discrete spaces via deterministic encoders, but relies on a smooth approximation instead of using the straight-through gradient estimator.

Works on *representation learning* for RL (Gelada et al., 2019; Castro et al., 2021; Zhang et al., 2021; Zang et al., 2022) consider bisimulation metrics to optimize the representation quality, and aim at learning (continuous) representations which capture bisimulation, so that two states close in the representation are guaranteed to provide close and relevant information to optimize the performance of the controller. In particular, as in our work, *DeepMDPs* (Gelada et al., 2019) are learned by optimizing *local losses*, by assuming a deterministic MDP and without verifiable confidence measurement.

## 2 BACKGROUND

In the following, we write $\Delta(\mathcal{X})$ for the set of measures over (complete, separable metric space) $\mathcal{X}$.

**Markov decision processes** (MDPs) are tuples $\mathcal{M} = \langle \mathcal{S}, \mathcal{A}, \mathbf{P}, \mathcal{R}, \ell, \mathbf{AP}, s_I \rangle$ where $\mathcal{S}$ is a set of *states*; $\mathcal{A}$, a set of *actions*; $\mathbf{P} \colon \mathcal{S} \times \mathcal{A} \to \Delta(\mathcal{S})$, a *probability transition function* that maps the current state and action to a *distribution* over the next states; $\mathcal{R} \colon \mathcal{S} \times \mathcal{A} \to \mathbb{R}$, a *reward function*; $\ell \colon \mathcal{S} \to 2^{\mathbf{AP}}$, a *labeling function* over a set of atomic propositions $\mathbf{AP}$; and $s_I \in \mathcal{S}$, the *initial state*. If $|\mathcal{A}| = 1$, $\mathcal{M}$ is a fully stochastic process called a *Markov chain* (MC). We write $\mathcal{M}_s$ for the MDP obtained when replacing the initial state of $\mathcal{M}$ by $s \in \mathcal{S}$. An agent interacting in $\mathcal{M}$ produces *trajectories*, i.e., sequences of states and actions $\tau = \langle s_{0:T}, a_{0:T-1} \rangle$ where $s_0 = s_I$ and $s_{t+1} \sim \mathbf{P}(\cdot \mid s_t, a_t)$ for $t < T$. The set of infinite trajectories of $\mathcal{M}$ is *Traj*. We assume $\mathbf{AP}$ and

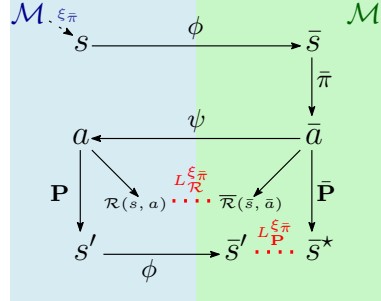 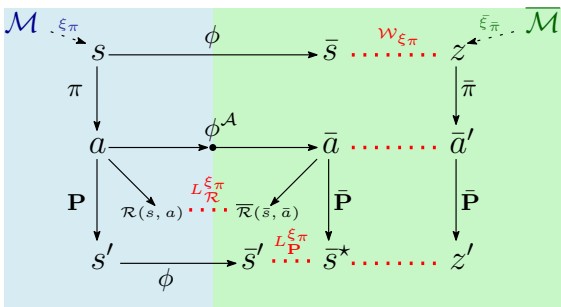

(a) Execution of the latent policy $\bar{\pi}$ in the original and latent MDPs, and local losses.

(b) Parallel execution of the original RL policy $\pi$ in the original and latent MDPs, local losses, and steady-state regularizer.

Figure 1: Latent flows: arrows represent (stochastic) mappings, the original (resp. latent) state-action space is spread along the blue (resp. green) area, and distances are depicted in red. Distilling $\pi$ into $\bar{\pi}$ via flow (b) by minimizing $\mathcal{W}_{\xi_\pi}$ allows closing the gap between flows (a) and (b).

labels being respectively one-hot and binary encoded. Given $\mathsf{T} \subseteq \mathbf{AP}$, we write $s \models \mathsf{T}$ if $s$ is labeled with $\mathsf{T}$, i.e., $\ell(s) \cap \mathsf{T} \neq \varnothing$, and $s \models \neg\mathsf{T}$ for $s \not\models \mathsf{T}$. We refer to MDPs with continuous state or action spaces as *continuous MDPs*. In that case, we assume $\mathcal{S}$ and $\mathcal{A}$ are complete separable metric spaces equipped with a Borel $\sigma$-algebra, and $\ell^{-1}(\mathsf{T})$ is Borel-measurable for any $\mathsf{T} \subseteq \mathbf{AP}$.

**Policies and stationary distributions.** A *(memoryless) policy* $\pi \colon \mathcal{S} \to \Delta(\mathcal{A})$ prescribes which action to choose at each step of the interaction. The set of memoryless policies of $\mathcal{M}$ is $\Pi$. The MDP $\mathcal{M}$ and $\pi \in \Pi$ induce an MC $\mathcal{M}_\pi$ with unique probability measure $\mathbb{P}_\pi^{\mathcal{M}}$ on the Borel $\sigma$-algebra over measurable subsets $\varphi \subseteq \mathit{Traj}$ (Puterman, 1994). We drop the superscript when the context is clear. Define $\xi_\pi^t(s' \mid s) = \mathbb{P}_\pi^{\mathcal{M}_s}(\{s_{0:\infty}, a_{0:\infty} \mid s_t = s'\})$ as the distribution giving the probability of being in each state of $\mathcal{M}_s$ after $t$ steps. $B \subseteq \mathcal{S}$ is a *bottom strongly connected component* (BSCC) of $\mathcal{M}_\pi$ if (i) $B$ is a maximal subset satisfying $\xi_\pi^t(s' \mid s) > 0$ for any $s, s' \in B$ and some $t \geqslant 0$, and (ii) $\mathbb{E}_{a \sim \pi(\cdot \mid s)} \mathbf{P}(B \mid s, a) = 1$ for all $s \in \mathcal{S}$. The unique stationary distribution of $B$ is $\xi_\pi \in \Delta(B)$. We write $s, a \sim \xi_\pi$ for sampling $s$ from $\xi_\pi$ then $a$ from $\pi$. An MDP $\mathcal{M}$ is *ergodic* if for all $\pi \in \Pi$, the state space of $\mathcal{M}_\pi$ consists of a unique aperiodic BSCC with $\xi_\pi = \lim_{t \to \infty} \xi_\pi^t(\cdot \mid s)$ for all $s \in \mathcal{S}$.

**Value objectives.** Given $\pi \in \Pi$, the *value* of a state $s \in \mathcal{S}$ is the expected value of a random variable obtained by running $\pi$ from $s$. For a discount factor $\gamma \in [0, 1]$, we consider the following objectives. (i) *Discounted return*: we write $V_\pi(s) = \mathbb{E}_\pi^{\mathcal{M}_s}\left[\sum_{t=0}^{\infty} \gamma^t \mathcal{R}(s_t, a_t)\right]$ for the expected discounted rewards accumulated along trajectories. The typical goal of an RL agent is to learn a policy $\pi^\star$ that maximizes $V_{\pi^\star}(s_I)$ through interactions with the (unknown) MDP; (ii) *Reachability*: let $\mathsf{C}, \mathsf{T} \subseteq \mathbf{AP}$, the *(constrained) reachability* event is $\mathsf{C}\,\mathcal{U}\,\mathsf{T} = \{\, s_{0:\infty}, a_{0:\infty} \mid \exists i \in \mathbb{N}, \forall j < i, s_j \models \mathsf{C} \wedge s_i \models \mathsf{T} \,\} \subseteq \mathit{Traj}$. We write $V_\pi^\varphi(s) = \mathbb{E}_\pi^{\mathcal{M}_s}\left[\gamma^{t^\star} \mathbf{1}_{\langle s_{0:\infty}, a_{0:\infty}\rangle \in \varphi}\right]$ for the *discounted probability of satisfying* $\varphi = \mathsf{C}\,\mathcal{U}\,\mathsf{T}$, where $t^\star$ is the length of the shortest trajectory prefix that allows satisfying $\varphi$. Intuitively, this denotes the discounted return of remaining in a region of the MDP where states are labeled with $\mathsf{C}$, until visiting *for the first time* a *goal state* labeled with $\mathsf{T}$, and the return is the binary reward signal capturing this event. *Safety* w.r.t. failure states $\mathsf{C}$ can be expressed as the safety-constrained reachability to a destination $\mathsf{T}$ through $\neg\mathsf{C}\,\mathcal{U}\,\mathsf{T}$. Notice that $V_\pi^\varphi(s) = \mathbb{P}_\pi^{\mathcal{M}_s}(\varphi)$ when $\gamma = 1$.

**Latent MDP.** Given the original (continuous, possibly unknown) environment model $\mathcal{M}$, a *latent space model* is another (smaller, explicit) MDP $\overline{\mathcal{M}} = \langle \overline{\mathcal{S}}, \overline{\mathcal{A}}, \overline{\mathbf{P}}, \overline{\mathcal{R}}, \bar{\ell}, \mathbf{AP}, \bar{s}_I \rangle$ with state-action space linked to the original one via state and action *embedding functions*: $\phi \colon \mathcal{S} \to \overline{\mathcal{S}}$ and $\psi \colon \overline{\mathcal{S}} \times \overline{\mathcal{A}} \to \mathcal{A}$. We refer to $\langle \mathcal{M}, \phi, \psi \rangle$ as a *latent space model* of $\mathcal{M}$ and $\overline{\mathcal{M}}$ as its *latent MDP*. Our goal is to learn $\langle \overline{\mathcal{M}}, \phi, \psi \rangle$ by optimizing an *equivalence criterion* between the two models. We assume that $d_{\overline{\mathcal{S}}}$ is a metric on $\overline{\mathcal{S}}$, and write $\overline{\Pi}$ for the set of policies of $\overline{\mathcal{M}}$ and $\overline{V}_{\bar{\pi}}$ for the values of running $\bar{\pi} \in \overline{\Pi}$ in $\overline{\mathcal{M}}$.

*Remark* 1 (Latent flow). The latent policy $\bar{\pi}$ can be seen as a policy in $\mathcal{M}$ (cf. Fig. 1a): states passed to $\bar{\pi}$ are first embedded with $\phi$ to the latent space, then the actions produced by $\bar{\pi}$ are executed via $\psi$ in the original environment. Let $s \in \mathcal{S}$, we write $\bar{a} \sim \bar{\pi}(\cdot \mid s)$ for $\bar{\pi}(\cdot \mid \phi(s))$, then the reward and next state are respectively given by $\mathcal{R}(s, \bar{a}) = \mathcal{R}(s, \psi(\phi(s), \bar{a}))$ and $s' \sim \mathbf{P}(\cdot \mid s, \bar{a}) = \mathbf{P}(\cdot \mid s, \psi(\phi(s), \bar{a}))$.

**Local losses** allow quantifying the distance between the original and latent reward/transition functions *in the local setting*, i.e., under a given state-action distribution $\xi \in \Delta(\mathcal{S} \times \overline{\mathcal{A}})$:

$$L_{\mathcal{R}}^{\xi} = \underset{s,\bar{a} \sim \xi}{\mathbb{E}} \left| \mathcal{R}(s,\bar{a}) - \overline{\mathcal{R}}(\phi(s),\bar{a}) \right|, \qquad L_{\mathbf{P}}^{\xi} = \underset{s,\bar{a} \sim \xi}{\mathbb{E}} D\big(\phi\mathbf{P}(\cdot \mid s,\bar{a}), \overline{\mathbf{P}}(\cdot \mid \phi(s),\bar{a})\big)$$

where $\phi\mathbf{P}(\cdot \mid s,\bar{a})$ is the distribution of drawing $s' \sim \mathbf{P}(\cdot \mid s,\bar{a})$ then embedding $\bar{s}' = \phi(s')$, and $D$ is a discrepancy measure. Fig 1a depicts the losses when states and actions are drawn from a stationary distribution $\xi_{\bar{\pi}}$ resulting from running $\bar{\pi} \in \overline{\Pi}$ in $\mathcal{M}$. In this work, we focus on the case where $D$ is the *Wasserstein distance* $W_{d_{\overline{\mathcal{S}}}}$: given two distributions $P, Q$ over a measurable set $\mathcal{X}$ equipped with a metric $d$, $W_d$ is the solution of the *optimal transport* (OT) from $P$ to $Q$, i.e., the minimum cost of changing $P$ into $Q$ (Villani, 2009): $W_d(P,Q) = \inf_{\lambda \in \Lambda(P,Q)} \mathbb{E}_{x,y \sim \lambda} d(x,y)$, $\Lambda(P,Q)$ being the set of all *couplings* of $P$ and $Q$. The *Kantorovich duality* yields $W_d(P,Q) = \sup_{f \in \mathcal{F}_d} \mathbb{E}_{x \sim P} f(x) - \mathbb{E}_{x \sim Q} f(y)$ where $\mathcal{F}_d$ is the set of 1-Lipschiz functions. Local losses are related to a well-established *behavioral* equivalence between transition systems, called *bisimulation*.

**Bisimulation.** A *bisimulation* $\mathcal{B}$ on $\mathcal{M}$ is a behavioral equivalence between states $s_1, s_2 \in \mathcal{S}$ so that, $s_1 \mathcal{B} s_2$ iff (i) $\mathbf{P}(T \mid s_1, a) = \mathbf{P}(T \mid s_2, a)$, (ii) $\ell(s_1) = \ell(s_2)$, and (iii) $\mathcal{R}(s_1, a) = \mathcal{R}(s_2, a)$ for each action $a \in \mathcal{A}$ and (Borel measurable) equivalence class $T \in \mathcal{S}/\mathcal{B}$. Properties of bisimulation include trajectory and value equivalence (Larsen & Skou, 1989; Givan et al., 2003). Requirements (ii) and (iii) can be respectively relaxed depending on whether we focus only on behaviors formalized through $\mathbf{AP}$ or rewards. The relation can be extended to compare two MDPs (e.g., $\mathcal{M}$ and $\overline{\mathcal{M}}$) by considering the disjoint union of their state space. We denote the largest bisimulation relation by $\sim$.

Characterized by a logical family of functional expressions derived from a logic $\mathcal{L}$, *bisimulation pseudometrics* (Desharnais et al., 2004) generalize the notion of bisimilariy. More specifically, given a policy $\pi \in \Pi$, we consider a family $\mathcal{F}$ of real-valued functions parameterized by a discount factor $\gamma$ and defining the semantics of $\mathcal{L}$ in $\mathcal{M}_\pi$. Such functional expressions allow to formalize discounted properties such as reachability, safety, as well as general $\omega$-regular specifications (Chatterjee et al., 2010) and may include rewards as well (Ferns et al., 2014). The pseudometric $\tilde{d}_\pi$ is defined as *the largest behavioral difference* $\tilde{d}_\pi(s_1, s_2) = \sup_{f \in \mathcal{F}} |f(s_1) - f(s_2)|$, and *its kernel is bisimilarity*: $\tilde{d}_\pi(s_1, s_2) = 0$ iff $s_1 \sim s_2$. In particular, *value functions are Lipschitz-continuous w.r.t.* $\tilde{d}_\pi$: $|V_\pi^\cdot(s_1) - V_\pi^\cdot(s_2)| \leqslant K \tilde{d}_\pi(s_1, s_2)$, where $K$ is $1/(1-\gamma)$ if rewards are included in $\mathcal{F}$ and $1$ otherwise. To ensure the upcoming bisimulation guarantees, we make the following assumptions:

**Assumption 2.1.** *MDP $\mathcal{M}$ is ergodic, $\mathrm{Im}(\mathcal{R})$ is a bounded space scaled in $[-1/2, 1/2]$, and the embedding function preserves the labels, i.e., $\phi(s) = \bar{s} \implies \ell(s) = \bar{\ell}(\bar{s})$ for $s \in \mathcal{S}, \bar{s} \in \overline{\mathcal{S}}$.*

Note that the ergodicity assumption is compliant with episodic RL and a wide range of continuous learning tasks (see Huang 2020; Delgrange et al. 2022 for detailed discussions on this setting).

**Bisimulation bounds (Delgrange et al., 2022).** $\mathcal{M}$ being set over continuous spaces with possibly unknown dynamics, evaluating $\tilde{d}$ can turn out to be particularly arduous, if not intractable. A solution is to evaluate the original and latent model bisimilarity via local losses: fix $\bar{\pi} \in \overline{\Pi}$, assume $\overline{\mathcal{M}}$ is discrete, then given the induced stationary distribution $\xi_{\bar{\pi}}$ in $\mathcal{M}$, let $s_1, s_2 \in \mathcal{S}$ with $\phi(s_1) = \phi(s_2)$:

$$\underset{s \sim \xi_{\bar{\pi}}}{\mathbb{E}} \tilde{d}_{\bar{\pi}}(s, \phi(s)) \leqslant \frac{L_{\mathcal{R}}^{\xi_{\bar{\pi}}} + \gamma L_{\mathbf{P}}^{\xi_{\bar{\pi}}}}{1 - \gamma}, \quad \tilde{d}_{\bar{\pi}}(s_1, s_2) \leqslant \left( \frac{L_{\mathcal{R}}^{\xi_{\bar{\pi}}} + \gamma L_{\mathbf{P}}^{\xi_{\bar{\pi}}}}{1 - \gamma} \right) \big(\xi_{\bar{\pi}}^{-1}(s_1) + \xi_{\bar{\pi}}^{-1}(s_2)\big). \quad (1)$$

The two inequalities guarantee respectively the *quality of the abstraction* and *representation*: when local losses are small, (i) states and their embedding are bisimilarly close in average, and (ii) all states sharing the same discrete representation are bisimilarly close. The local losses and related bounds can be efficiently PAC-estimated. Our goal is to learn a latent model where the behaviors of the agent executing $\bar{\pi}$ can be formally verified, and the bounds offer a confidence metric allowing to lift the guarantees obtained this way back to the original model $\mathcal{M}$, when the latter operates under $\bar{\pi}$. We show in the following how to learn a latent space model by optimizing the aforementioned bounds, and distill policies $\pi \in \Pi$ obtained via *any* RL technique to a latent policy $\bar{\pi} \in \overline{\Pi}$.

## 3 Wasserstein Auto-encoded MDPs

Fix $\overline{\mathcal{M}}_\theta = \langle \overline{\mathcal{S}}, \overline{\mathcal{A}}, \overline{\mathbf{P}}_\theta, \overline{\mathcal{R}}_\theta, \bar{\ell}, \mathbf{AP}, \bar{s}_I \rangle$ and $\langle \overline{\mathcal{M}}_\theta, \phi_\iota, \psi_\theta \rangle$ as a latent space model of $\mathcal{M}$ parameterized by $\iota$ and $\theta$. Our method relies on learning a *behavioral model* $\xi_\theta$ of $\mathcal{M}$ from which we can

retrieve the latent space model and distill $\pi$. This can be achieved via the minimization of a suitable discrepancy between $\xi_\theta$ and $\mathcal{M}_\pi$. VAE-MDPs optimize a lower bound on the likelihood of the dynamics of $\mathcal{M}_\pi$ using the *Kullback-Leibler divergence*, yielding (i) $\overline{\mathcal{M}}_\theta$, (ii) a distillation $\bar{\pi}_\theta$ of $\pi$, and (iii) $\phi_\iota$ and $\psi_\theta$. Local losses are not directly minimized, but rather variational proxies that do not offer theoretical guarantees during the learning process. To control the local losses minimization and exploit their theoretical guarantees, we present a novel autoencoder that incorporates them in its objective, derived from the OT. Proofs of the claims made in this Section are provided in Appendix A.

## 3.1 THE OBJECTIVE FUNCTION

Assume that $\mathcal{S}$, $\mathcal{A}$, and $\text{Im}(\mathcal{R})$ are respectively equipped with metrics $d_\mathcal{S}$, $d_\mathcal{A}$, and $d_\mathcal{R}$, we define the *raw transition distance metric* $\vec{d}$ as the component-wise sum of distances between states, actions, and rewards occurring of along transitions: $\vec{d}(\langle s_1, a_1, r_1, s_1' \rangle, \langle s_2, a_2, r_2, s_2' \rangle) = d_\mathcal{S}(s_1, s_2) + d_\mathcal{A}(a_1, a_2) + d_\mathcal{R}(r_1, r_2) + d_\mathcal{S}(s_1', s_2')$. Given Assumption 2.1, we consider the OT between *local* distributions, where traces are drawn from episodic RL processes or infinite interactions (we show in Appendix A.1 that considering the OT between trace-based distributions in the limit amounts to reasoning about stationary distributions). Our goal is to minimize $W_{\vec{d}}(\xi_\pi, \xi_\theta)$ so that

$$\xi_\theta(s, a, r, s') = \int_{\overline{\mathcal{S}} \times \overline{\mathcal{A}} \times \overline{\mathcal{S}}} P_\theta(s, a, r, s' \mid \bar{s}, \bar{a}, \bar{s}') \, d\bar{\xi}_{\bar{\pi}_\theta}(\bar{s}, \bar{a}, \bar{s}'), \tag{2}$$

where $P_\theta$ is a transition decoder and $\bar{\xi}_{\bar{\pi}_\theta}$ denotes the stationary distribution of the latent model $\overline{\mathcal{M}}_\theta$. As proved by Bousquet et al. (2017), this model allows to derive a simpler form of the OT: instead of finding the optimal coupling of (i) the stationary distribution $\xi_\pi$ of $\mathcal{M}_\pi$ and (ii) the behavioral model $\xi_\theta$, in the primal definition of $W_{\vec{d}}(\xi_\pi, \xi_\theta)$, it is sufficient to find an encoder $q$ whose marginal is given by $Q(\bar{s}, \bar{a}, \bar{s}') = \mathbb{E}_{s,a,s' \sim \xi_\pi} q(\bar{s}, \bar{a}, \bar{s}' \mid s, a, s')$ and identical to $\xi_\pi$. This is summarized in the following Theorem, yielding a particular case of *Wasserstein-autoencoder* Tolstikhin et al. (2018):

**Theorem 3.1.** *Let $\xi_\theta$ and $P_\theta$ be respectively a behavioral model and transition decoder as defined in Eq. 2, $\mathcal{G}_\theta \colon \overline{\mathcal{S}} \to \mathcal{S}$ be a state-wise decoder, and $\psi_\theta$ be an action embedding function. Assume $P_\theta$ is deterministic with Dirac function $G_\theta(\bar{s}, \bar{a}, \bar{s}') = \langle \mathcal{G}_\theta(\bar{s}), \psi_\theta(\bar{s}, \bar{a}), \overline{\mathcal{R}}_\theta(\bar{s}, \bar{a}), \mathcal{G}_\theta(\bar{s}') \rangle$, then*

$$W_{\vec{d}}(\xi_\pi, \xi_\theta) = \inf_{q \colon Q = \bar{\xi}_{\bar{\pi}_\theta}} \mathbb{E}_{s,a,r,s' \sim \xi_\pi} \mathbb{E}_{\bar{s}, \bar{a}, \bar{s}' \sim q(\cdot \mid s,a,s')} \vec{d}(\langle s, a, r, s' \rangle, G_\theta(\bar{s}, \bar{a}, \bar{s}')).$$

Henceforth, fix $\phi_\iota \colon \mathcal{S} \to \overline{\mathcal{S}}$ and $\phi_\iota^\mathcal{A} \colon \overline{\mathcal{S}} \times \mathcal{A} \to \Delta(\overline{\mathcal{A}})$ as parameterized state and action encoders with $\phi_\iota(\bar{s}, \bar{a}, \bar{s}' \mid s, a, s') = \mathbf{1}_{\phi_\iota(s) = \bar{s}} \cdot \phi_\iota^\mathcal{A}(\bar{a} \mid \bar{s}, a) \cdot \mathbf{1}_{\phi_\iota(s') = \bar{s}'}$, and define the marginal encoder as $Q_\iota = \mathbb{E}_{s,a,s' \sim \xi_\pi} \phi_\iota(\cdot \mid s, a, s')$. Training the model components can be achieved via the objective:

$$\min_{\iota, \theta} \mathbb{E}_{s,a,r,s' \sim \xi_\pi} \mathbb{E}_{\bar{s}, \bar{a}, \bar{s}' \sim \phi_\iota(\cdot \mid s,a,s')} \vec{d}(\langle s, a, r, s' \rangle, G_\theta(\bar{s}, \bar{a}, \bar{s}')) + \beta \cdot D(Q_\iota, \bar{\xi}_{\bar{\pi}_\theta}),$$

where $D$ is an arbitrary discrepancy metric and $\beta > 0$ a hyperparameter. Intuitively, the encoder $\phi_\iota$ can be learned by enforcing its marginal distribution $Q_\iota$ to match $\bar{\xi}_{\bar{\pi}_\theta}$ through this discrepancy.

*Remark* 2. If $\mathcal{M}$ has a discrete action space, then learning $\overline{\mathcal{A}}$ is not necessary. We can set $\overline{\mathcal{A}} = \mathcal{A}$ using identity functions for the action encoder and decoder (details in Appendix A.2).

When $\pi$ is executed in $\mathcal{M}$, observe that its *parallel execution* in $\overline{\mathcal{M}}_\theta$ is enabled by the action encoder $\phi_\iota^\mathcal{A}$: given an original state $s \in \mathcal{S}$, $\pi$ first prescribes the action $a \sim \pi(\cdot \mid s)$, which is then embedded in the latent space via $\bar{a} \sim \phi_\iota^\mathcal{A}(\cdot \mid \phi_\iota(s), a)$ (cf. Fig. 1b). This parallel execution, along with setting $D$ to $W_{\vec{d}}$, yield an upper bound on the latent regularization, compliant with the bisimulation bounds. A two-fold regularizer is obtained thereby, defining the foundations of our objective function:

**Lemma 3.2.** *Define $\mathcal{T}(\bar{s}, \bar{a}, \bar{s}') = \mathbb{E}_{s,a \sim \xi_\pi}[\mathbf{1}_{\phi_\iota(s) = \bar{s}} \cdot \phi_\iota^\mathcal{A}(\bar{a} \mid \bar{s}, a) \cdot \overline{\mathbf{P}}_\theta(\bar{s}' \mid \bar{s}, \bar{a})]$ as the distribution of drawing state-action pairs from interacting with $\mathcal{M}$, embedding them to the latent spaces, and finally letting them transition to their successor state in $\overline{\mathcal{M}}_\theta$. Then, $W_{\vec{d}}(Q_\iota, \bar{\xi}_{\bar{\pi}_\theta}) \leqslant W_{\vec{d}}(\bar{\xi}_{\bar{\pi}_\theta}, \mathcal{T}) + L_\mathbf{P}^{\xi_\pi}$.*

We therefore define the W²AE-MDP (*Wasserstein-Wasserstein auto-encoded MDP*) objective as:

$$\min_{\iota, \theta} \mathbb{E}_{\substack{s,a,s' \sim \xi_\pi \\ \bar{s}, \bar{a}, \bar{s}' \sim \phi_\iota(\cdot \mid s,a,s')}} [d_\mathcal{S}(s, \mathcal{G}_\theta(\bar{s})) + d_\mathcal{A}(a, \psi_\theta(\bar{s}, \bar{a})) + d_\mathcal{S}(s', \mathcal{G}_\theta(\bar{s}'))] + L_\mathcal{R}^{\xi_\pi} + \beta \cdot (\mathcal{W}_{\xi_\pi} + L_\mathbf{P}^{\xi_\pi}),$$

---

**Algorithm 1:** Wasserstein$^2$ Auto-Encoded MDP

---

**Input:** batch size $N$, max. step $T$, no. of regularizer updates $m$, penalty coefficient $\delta > 0$

**for** $t = 1$ *to* $T$ **do**

    **for** $i = 1$ *to* $N$ **do**

        Sample a transition $s_i, a_i, r_i, s'_i$ from the original environment via $\xi_\pi$

        Embed the transition into the latent space by drawing $\bar{s}_i, \bar{a}_i, \bar{s}'_i$ from $\phi_\iota(\cdot \mid s_i, a_i, s'_i)$

        Make the latent space model transition to the next latent state: $\bar{s}_i^\star \sim \overline{\mathbf{P}}_\theta(\cdot \mid \bar{s}_i, \bar{a}_i)$

        Sample a latent transition from $\bar{\xi}_{\bar{\pi}_\theta}$: $z_i \sim \bar{\xi}_{\bar{\pi}_\theta}, \bar{a}'_i \sim \bar{\pi}_\theta(\cdot \mid z_i)$, and $z'_i \sim \overline{\mathbf{P}}_\theta(\cdot \mid z_i, \bar{a}'_i)$

    $\mathcal{W} \leftarrow \sum_{i=1}^N \varphi_\omega^\xi(\bar{s}_i, \bar{a}_i, \bar{s}_i^\star) - \varphi_\omega^\xi(z_i, \bar{a}'_i, z'_i) + \varphi_\omega^{\mathbf{P}}(s_i, a_i, \bar{s}_i, \bar{a}_i, \bar{s}'_i) - \varphi_\omega^{\mathbf{P}}(s_i, a_i, \bar{s}_i, \bar{a}_i, \bar{s}_i^\star)$

    $P \leftarrow \sum_{i=1}^N \mathrm{GP}(\varphi_\omega^\xi, \langle\bar{s}_i, \bar{a}_i, \bar{s}_i^\star\rangle, \langle z_i, \bar{a}'_i, z'_i\rangle) + \mathrm{GP}(\boldsymbol{x} \mapsto \varphi_\omega^{\mathbf{P}}(s_i, a_i, \bar{s}_i, \bar{a}_i, \boldsymbol{x}), \bar{s}'_i, \bar{s}_i^\star)$

    Update the Lipschitz networks parameters $\omega$ by ascending $^1/_N \cdot (\beta\,\mathcal{W} - \delta\,P)$

    **if** $t \bmod m = 0$ **then**

        $\mathcal{L} \leftarrow \sum_{i=1}^N d_\mathcal{S}(s_i, \mathcal{G}_\theta(\bar{s}_i)) + d_\mathcal{A}(a_i, \psi_\theta(\bar{s}_i, \bar{a}_i)) + d_\mathcal{R}(r_i, \overline{\mathcal{R}}_\theta(\bar{s}_i, \bar{a}_i)) + d_\mathcal{S}(s'_i, \mathcal{G}_\theta(\bar{s}'_i))$

        Update the latent space model parameters $\langle\iota, \theta\rangle$ by descending $^1/_N \cdot (\mathcal{L} + \beta\,\mathcal{W})$

**function** $\mathrm{GP}(\varphi_\omega, \boldsymbol{x}, \boldsymbol{y})$          $\rhd$ **Gradient penalty** for $\varphi_\omega \colon \mathbb{R}^n \to \mathbb{R}$ and $\boldsymbol{x}, \boldsymbol{y} \in \mathbb{R}^n$

    $\epsilon \sim U(0, 1); \tilde{\boldsymbol{x}} \leftarrow \epsilon\boldsymbol{x} + (1 - \epsilon)\boldsymbol{y}$          $\rhd$ random noise; straight lines between $\boldsymbol{x}$ and $\boldsymbol{y}$

    **return** $(\|\nabla_{\tilde{\boldsymbol{x}}}\,\varphi_\omega(\tilde{\boldsymbol{x}})\| - 1)^2$

---

where $\mathcal{W}_{\xi_\pi} = W_{\vec{d}}(\mathcal{T}, \bar{\xi}_{\bar{\pi}_\theta})$ and $L_{\mathbf{P}}^{\xi_\pi}$ are respectively called *steady-state* and *transition* regularizers. The former allows to quantify the distance between the stationary distributions respectively induced by $\pi$ in $\mathcal{M}$ and $\bar{\pi}_\theta$ in $\overline{\mathcal{M}}_\theta$, further enabling the distillation. The latter allows to learn the latent dynamics. Note that $L_{\mathcal{R}}^{\xi_\pi}$ and $L_{\mathbf{P}}^{\xi_\pi}$ — set over $\xi_\pi$ instead of $\xi_{\bar{\pi}_\theta}$ — are not sufficient to ensure the bisimulation bounds (Eq. 1): running $\pi$ in $\overline{\mathcal{M}}_\theta$ depends on the parallel execution of $\pi$ in the original model, which does not permit its (conventional) verification. Breaking this dependency is enabled by learning the distillation $\bar{\pi}_\theta$ through $\mathcal{W}_{\xi_\pi}$, as shown in Fig. 1b: minimizing $\mathcal{W}_{\xi_\pi}$ allows to make $\xi_\pi$ and $\bar{\xi}_{\bar{\pi}_\theta}$ closer together, further bridging the gap of the discrepancy between $\pi$ and $\bar{\pi}_\theta$. At any time, recovering the local losses along with the linked bisimulation bounds in the objective function of the W$^2$AE-MDP is allowed by considering the latent policy resulting from this distillation:

**Theorem 3.3.** *Assume that traces are generated by running a latent policy $\bar{\pi} \in \overline{\Pi}$ in the original environment and let $d_\mathcal{R}$ be the usual Euclidean distance, then the W$^2$AE-MDP objective is*

$$\min_{\iota, \theta} \mathbb{E}_{s, s' \sim \xi_{\bar{\pi}}} \left[ d_\mathcal{S}(s, \mathcal{G}_\theta(\phi_\iota(s))) + d_\mathcal{S}(s', \mathcal{G}_\theta(\phi_\iota(s'))) \right] + L_\mathcal{R}^{\xi_{\bar{\pi}}} + \beta \cdot (\mathcal{W}_{\xi_{\bar{\pi}}} + L_{\mathbf{P}}^{\xi_{\bar{\pi}}}).$$

**Optimizing the regularizers** is enabled by the dual form of the OT: we introduce two parameterized networks, $\varphi_\omega^\xi$ and $\varphi_\omega^{\mathbf{P}}$, constrained to be 1-Lipschitz and trained to attain the supremum of the dual:

$$\mathcal{W}_{\xi_\pi}(\omega) = \max_\omega \mathbb{E}_{s, a \sim \xi_\pi} \mathbb{E}_{\bar{a} \sim \phi_\iota^\mathcal{A}(\cdot \mid \phi_\iota(s), a)} \mathbb{E}_{\bar{s}^\star \sim \overline{\mathbf{P}}_\theta(\cdot \mid \phi_\iota(s), \bar{a})} \varphi_\omega^\xi(\phi_\iota(s), \bar{a}, \bar{s}^\star) - \mathbb{E}_{z, \bar{a}', z' \sim \bar{\xi}_{\bar{\pi}_\theta}} \varphi_\omega^\xi(z, \bar{a}', z')$$

$$L_{\mathbf{P}}^{\xi_\pi}(\omega) = \max_\omega \mathbb{E}_{s, a, s' \sim \xi_\pi} \mathbb{E}_{\bar{s}, \bar{a}, \bar{s}' \sim \phi_\iota(\cdot \mid s, a, s')} \left[ \varphi_\omega^{\mathbf{P}}(s, a, \bar{s}, \bar{a}, \bar{s}') - \mathbb{E}_{\bar{s}^\star \sim \overline{\mathbf{P}}_\theta(\cdot \mid \bar{s}, \bar{a})} \varphi_\omega^{\mathbf{P}}(s, a, \bar{s}, \bar{a}, \bar{s}^\star) \right]$$

Details to derive this tractable form of $L_{\mathbf{P}}^{\xi_\pi}(\omega)$ are in Appendix A.5. The networks are constrained via the gradient penalty approach of Gulrajani et al. (2017), leveraging that any differentiable function is 1-Lipschitz iff it has gradients with norm at most 1 everywhere (we show in Appendix A.6 this is still valid for relaxations of discrete spaces). The final learning process is presented in Algorithm 1.

## 3.2 DISCRETE LATENT SPACES

To enable the verification of latent models supported by the bisimulation guarantees of Eq. 1, we focus on the special case of *discrete latent space models*. Our approach relies on continuous relaxation of discrete random variables, regulated by some *temperature* parameter(s) $\lambda$: discrete random variables are retrieved as $\lambda \to 0$, which amounts to applying a rounding operator. For training, we use the

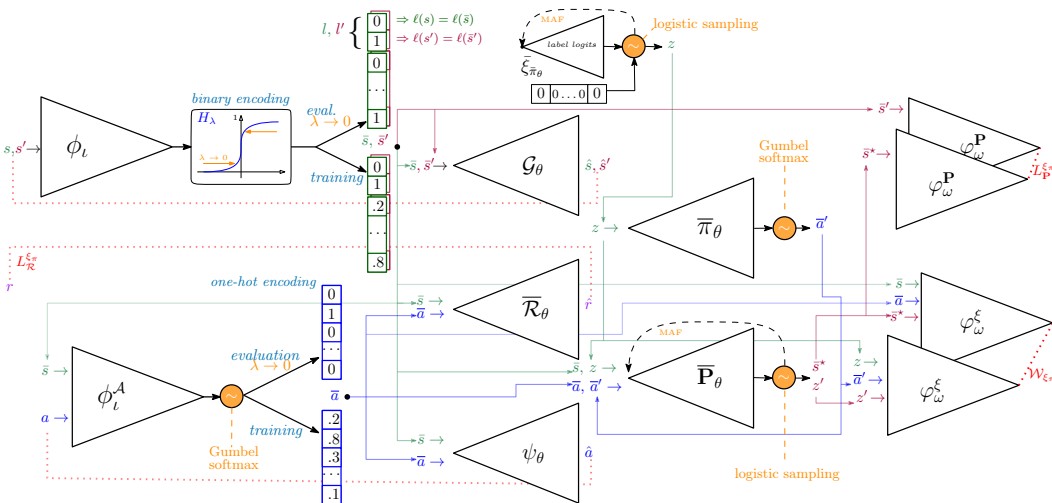

Figure 2: W²AE-MDP architecture. Distances are depicted by red dotted lines.

temperature-controlled relaxations to differentiate the objective and let the gradient flow through the network. When we deploy the latent policy in the environment and formally check the latent model, the zero-temperature limit is used. An overview of the approach is depticted in Fig. 2.

**State encoder.** We work with a *binary representation* of the latent states. First, this induces compact networks, able to deal with a large discrete space via a tractable number of parameter variables. But most importantly, this ensures that Assumption 2.1 is satisfied: let $n = \log_2 |\bar{\mathcal{S}}|$, we reserve $|\mathbf{AP}|$ bits in $\bar{\mathcal{S}}$ and each time $s \in \mathcal{S}$ is passed to $\phi_\iota$, $n - |\mathbf{AP}|$ bits are produced and concatenated with $\ell(s)$, ensuring a perfect reconstruction of the labels and further bisimulation bounds. To produce Bernoulli variables, $\phi_\iota$ deterministically maps $s$ to a latent code $\boldsymbol{z}$, passed to the Heaviside $H(\boldsymbol{z}) = \mathbf{1}_{\boldsymbol{z}>0}$. We train $\phi_\iota$ by using the smooth approximation $H_\lambda(\boldsymbol{z}) = \sigma(2\boldsymbol{z}/\lambda)$, satisfying $H = \lim_{\lambda\to 0} H_\lambda$.

**Latent distributions.** Besides the discontinuity of their latent image space, a major challenge of optimizing over discrete distributions is *sampling*, required to be a differentiable operation. We circumvent this by using *concrete distributions* (Jang et al., 2017; Maddison et al., 2017): the idea is to sample reparameterizable random variables from $\lambda$-parameterized distributions, and applying a differentiable, nonlinear operator in downstream. We use the *Gumbel softmax trick* to sample from distributions over (one-hot encoded) latent actions ($\phi_\iota^{\mathcal{A}}$, $\overline{\pi}_\theta$). For binary distributions ($\overline{\mathbf{P}}_\theta$, $\bar{\xi}_{\overline{\pi}_\theta}$), each relaxed Bernoulli with logit $\alpha$ is retrieved by drawing a logistic random variable located in $\alpha/\lambda$ and scaled to $1/\lambda$, then applying a sigmoid in downstream. We emphasize that this trick alone (as used by Corneil et al. 2018; Delgrange et al. 2022) is not sufficient: it yields independent Bernoullis, being too restrictive in general, which prevents from learning sound transition dynamics (cf. Example 1).

*Example* 1. Let $\overline{\mathcal{M}}$ be the discrete MC of Fig. 3. In one-hot, $\mathbf{AP} = \{goal : \langle 1, 0\rangle, unsafe : \langle 0, 1\rangle\}$. We assume that 3 bits are used for the (binary) state space, with $\bar{\mathcal{S}} = \{\bar{s}_0 : \langle 0, 0, 0\rangle, \bar{s}_1 : \langle 1, 0, 0\rangle, \bar{s}_2 : \langle 0, 1, 0\rangle, \bar{s}_3 : \langle 0, 1, 1\rangle\}$ (the two first bits are reserved for the labels). Considering each bit as being independent is not sufficient to learn $\overline{\mathbf{P}}$: the optimal estimation $\overline{\mathbf{P}}_{\theta^\star}(\cdot \mid \bar{s}_0)$ is in that case represented by the independent Bernoulli vector $\mathbf{b} = \langle 1/2, 1/2, 1/4\rangle$, giving the probability to go from $\bar{s}_0$ to each bit *independently*. This yields a poor estimation of

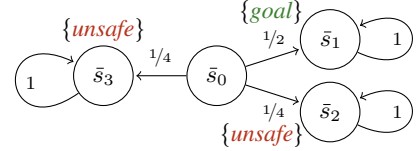

Figure 3: Markov Chain with four states; labels are drawn next to their state.

the actual transition function: $\overline{\mathbf{P}}_{\theta^\star}(\bar{s}_0 \mid \bar{s}_0) = (1-\mathbf{b}_1)\cdot(1-\mathbf{b}_2)\cdot(1-\mathbf{b}_3) = \overline{\mathbf{P}}_{\theta^\star}(\bar{s}_1 \mid \bar{s}_0) = \mathbf{b}_1\cdot(1-\mathbf{b}_2)\cdot(1-\mathbf{b}_3) = \overline{\mathbf{P}}_{\theta^\star}(\bar{s}_2 \mid \bar{s}_0) = (1-\mathbf{b}_1)\cdot\mathbf{b}_2\cdot(1-\mathbf{b}_3) = 3/16$, $\overline{\mathbf{P}}_{\theta^\star}(\bar{s}_3 \mid \bar{s}_0) = (1-\mathbf{b}_1)\cdot\mathbf{b}_2\cdot\mathbf{b}_3 = 1/16$.

We consider instead relaxed multivariate Bernoulli distributions by decomposing $P \in \Delta(\bar{\mathcal{S}})$ as a product of conditionals: $P(\bar{s}) = \prod_{i=1}^{n} P(\bar{s}_i \mid \bar{s}_{1: i-1})$ where $\bar{s}_i$ is the $i^{\text{th}}$ entry (bit) of $\bar{s}$. We learn

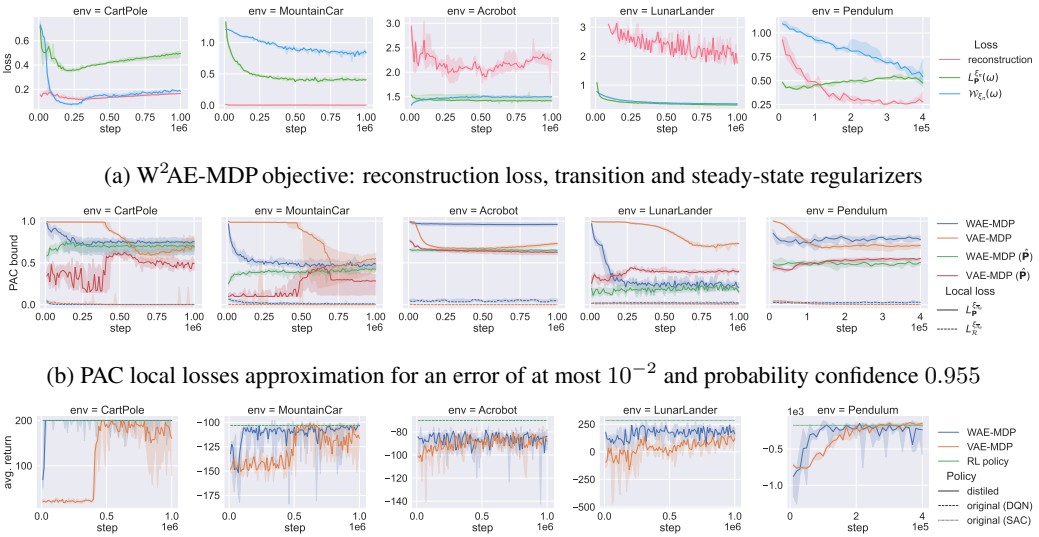

(a) $W^2$AE-MDP objective: reconstruction loss, transition and steady-state regularizers

(b) PAC local losses approximation for an error of at most $10^{-2}$ and probability confidence $0.955$

(c) Episode return obtained when executing the distilled policy in the original MDP (averaged over 30 episodes)

Figure 4: For each environment, we trained five different instances of the models with different random seeds: the solid line is the median and the shaded interval the interquartile range.

such distributions by introducing a *masked autoregressive flow* (MAF, Papamakarios et al. 2017) for relaxed Bernoullis via the recursion: $\bar{s}_i = \sigma(^{l_i + \alpha_i}/\lambda)$, where $l_i \sim \text{Logistic}(0, 1)$, $\alpha_i = f_i(\bar{s}_{1:\, i-1})$, and $f$ is a MADE (Germain et al., 2015), a feedforward network implementing the conditional output dependency on the inputs via a mask that only keeps the necessary connections to enforce the conditional property. We use this MAF to model $\overline{\mathbf{P}}_\theta$ and the dynamics related to the labels in $\bar{\xi}_{\overline{\pi}_\theta}$. We fix the logits of the remaining $n - |\mathbf{AP}|$ bits to $0$ to allow for a fairly distributed latent space.

## 4    EXPERIMENTS

We evaluate the quality of latent space models learned and policies distilled through $W^2$AE-MDPs . To do so, we first trained deep-RL policies (DQN, Mnih et al. 2015 on discrete, and SAC, Haarnoja et al. 2018 on continuous action spaces) for various OpenAI benchmarks (Brockman et al., 2016), which we then distill via our approach (Figure 4). We thus evaluate (a) the $W^2$AE-MDP training metrics, (b) the abstraction and representation quality via *PAC local losses upper bounds* (Delgrange et al., 2022), and (c) the distilled policy performance when deployed in the original environment. The confidence metrics and performance are compared with those of VAE-MDPs. Finally, we formally verify properties in the latent model. The exact setting to reproduce our results is in Appendix B.

**Learning metrics.** The objective (Fig. 4a) is a weighted sum of the reconstruction loss and the two Wasserstein regularizers. The choice of $\beta$ defines the optimization direction. In contrast to VAEs (cf. Appendix C), WAEs indeed naturally avoid posterior collapse (Tolstikhin et al., 2018), indicating that the latent space is consistently distributed. Optimizing the objective (Fig. 4a) effectively allows minimizing the local losses (Fig. 4b) and recovering the performance of the original policy (Fig. 4c).

**Local losses.** For V- and WAEs, we formally evaluate PAC upper bounds on $L_{\mathcal{R}}^{\xi_{\overline{\pi}_\theta}}$ and $L_{\mathbf{P}}^{\xi_{\overline{\pi}_\theta}}$ via the algorithm of Delgrange et al. (2022) (Fig 4b). The lower the local losses, the closer $\mathcal{M}$ and $\overline{\mathcal{M}}_\theta$ are in terms of behaviors induced by $\overline{\pi}_\theta$ (cf. Eq. 1). In VAEs, the losses are evaluated on a transition function $\hat{\mathbf{P}}$ obtained via frequency estimation of the latent transition dynamics (Delgrange et al., 2022), by reconstructing the transition model a posteriori and collecting data to estimate the transition probabilities (e.g., Bazille et al. 2020; Corneil et al. 2018). We thus also report the metrics for $\hat{\mathbf{P}}$. Our bounds quickly converge to close values in general for $\overline{\mathbf{P}}_\theta$ and $\hat{\mathbf{P}}$, whereas for VAEs, the convergence is slow and unstable, with $\hat{\mathbf{P}}$ offering better bounds. We emphasize that WAEs do not require this additional reconstruction step to obtain losses that can be leveraged to assess the

Table 1: Formal Verification of distilled policies. Values are computed for $\gamma = 0.99$ (lower is better).

| Environment | step ($10^5$) | $\mathcal{S}$ | $\mathcal{A}$ | $|\bar{\mathcal{S}}|$ | $|\bar{\mathcal{A}}|$ | $L_{\mathcal{R}}^{\xi_{\bar{\pi}_\theta}}$ (PAC) | $L_{\mathbf{P}}^{\xi_{\bar{\pi}_\theta}}$ (PAC) | $\|V_{\bar{\pi}_\theta}\|$ | $\bar{V}_{\bar{\pi}_\theta}^{\varphi}(\bar{s}_I)$ |
|---|---|---|---|---|---|---|---|---|---|
| CartPole | 1.2 | $\subseteq \mathbb{R}^4$ | $\{1,2\}$ | 512 | 2 | 0.00499653 | 0.399636 | 3.71213 | 0.0316655 |
| MountainCar | 2.32 | $\subseteq \mathbb{R}^2$ | $\{1,2\}$ | 1024 | 2 | 0.0141763 | 0.382323 | 2.83714 | 0 |
| Acrobot | 4.3 | $\subseteq \mathbb{R}^6$ | $\{1,2,3\}$ | 8192 | 3 | 0.0347698 | 0.649478 | 2.22006 | 0.0021911 |
| LunarLander | 3.2 | $\subseteq \mathbb{R}^8$ | $[-1,1]^2$ | 16384 | 3 | 0.0207205 | 0.131357 | 0.0372883 | 0.0702039 |
| Pendulum | 3.7 | $\subseteq \mathbb{R}^3$ | $[-2,2]$ | 8192 | 3 | 0.0266745 | 0.539508 | 4.33006 | 0.0348492 |

quality of the model, in contrast to VAEs, where learning $\overline{\mathbf{P}}_\theta$ was performed via overly restrictive distributions, leading to poor estimation in general (cf. Ex. 1). Finally, *when the distilled policies offer comparable performance* (Fig. 4c), our bounds are either close to or better than those of VAEs.

**Distillation.** The bisimulation guarantees (Eq. 1) are only valid for $\bar{\pi}_\theta$, the policy under which formal properties can be verified. It is crucial that $\bar{\pi}_\theta$ achieves performance close to $\pi$, the original one, when deployed in the RL environment. We evaluate the performance of $\bar{\pi}_\theta$ via the undiscounted episode return $\mathbf{R}_{\bar{\pi}_\theta}$ obtained by running $\bar{\pi}_\theta$ in the original model $\mathcal{M}$. We observe that $\mathbf{R}_{\bar{\pi}_\theta}$ approaches faster the original performance $\mathbf{R}_\pi$ for W- than VAEs: WAEs converge in a few steps for all environments, whereas the full learning budget is sometimes necessary with VAEs. The success in recovering the original performance emphasizes the representation quality guarantees (Eq. 1) induced by WAEs: when local losses are minimized, all original states that are embedded to the same representation are bisimilarly close. Distilling the policy over the new representation, albeit discrete and hence coarser, still achieves effective performance since $\phi_\iota$ keeps only what is important to preserve behaviors, and thus values. Furthermore, the distillation can remove some non-robustness obtained during RL: $\bar{\pi}_\theta$ prescribes the same actions for bisimilarly close states, whereas this is not necessarily the case for $\pi$.

**Formal verification.** To formally verify $\overline{\mathcal{M}}_\theta$, we implemented a *value iteration* (VI) engine, handling the neural network encoding of the latent space for discounted properties, which is one of the most popular algorithms for checking property probabilities in MDPs (e.g., Baier & Katoen 2008; Hensel et al. 2021; Kwiatkowska et al. 2022). We verify *time-to-failure* properties $\varphi$, often used to check the failure rate of a system (Pnueli, 1977) by measuring whether the agent fails *before the end of the episode*. Although simple, such properties highlight the applicability of our approach on reachability events, which are building blocks to verify MDPs (Baier & Katoen 2008; cf. Appendix B.7). In particular, we checked whether the agent reaches an unsafe position or angle (CartPole, LunarLander), does not reach its goal position (MountainCar, Acrobot), and does not reach and stay in a safe region of the system (Pendulum). Results are in Table 1: for each environment, we select the distilled policy which gives the best trade-off between performance (episode return) and abstraction quality (local losses). As extra confidence metric, we report the value difference $\|V_{\bar{\pi}_\theta}\| = |V_{\bar{\pi}_\theta}(s_I) - \bar{V}_{\bar{\pi}_\theta}(\bar{s}_I)|$ obtained by executing $\bar{\pi}_\theta$ in $\mathcal{M}$ and $\overline{\mathcal{M}}_\theta$ ($V_{\bar{\pi}_\theta}(\cdot)$ is averaged while $\bar{V}_{\bar{\pi}_\theta}(\cdot)$ is formally computed).

## 5 CONCLUSION

We presented WAE-MDPs, a framework for learning formally verifiable distillations of RL policies with bisimulation guarantees. The latter, along with the learned abstraction of the unknown continuous environment to a discrete model, enables the verification. Our method overcomes the limitations of VAE-MDPs and our results show that it outperforms the latter in terms of learning speed, model quality, and performance, in addition to being supported by stronger learning guarantees. As mentioned by Delgrange et al. (2022), distillation failure reveals the lack of robustness of original RL policies. In particular, we found that distilling highly noise-sensitive RL policies (such as robotics simulations, e.g., Todorov et al. 2012) is laborious, even though the result remains formally verifiable.

We demonstrated the feasibility of our approach through the verification of reachability objectives, which are building blocks for stochastic model-checking (Baier & Katoen, 2008). Besides the scope of this work, the verification of general discounted $\omega$-regular properties is theoretically allowed in our model via the rechability to components of standard constructions based on automata products (e.g., Baier et al. 2016; Sickert et al. 2016), and discounted games algorithms (Chatterjee et al., 2010). Beyond distillation, our results, supported by Thm. 3.3, suggest that our WAE-MDP can be used as a *general latent space learner* for RL, further opening possibilities to combine RL and formal methods *online* when no formal model is a priori known, and address this way safety in RL with guarantees.

## REPRODUCIBILITY STATEMENT

We referenced in the main text the Appendix parts presenting the proofs or additional details of every claim, Assumption, Lemma, and Theorem occurring in the paper. In addition, Appendix B is dedicated to the presentation of the setup, hyperparameters, and other extra details required for reproducing the results of Section 4. We provide the source code of the implementation of our approach in Supplementary material [1], and we also provide the models saved during training that we used for model checking (i.e., reproducing the results of Table 1). Additionally, we present in a notebook (`evaluation.html`) videos demonstrating how our distilled policies behave in each environment, and code snippets showing how we formally verified the policies.

## ACKNOWLEDGMENTS

This research received funding from the Flemish Government (AI Research Program) and was supported by the DESCARTES iBOF project. G.A. Perez is also supported by the Belgian FWO "SAILor" project (G030020N). We thank Raphael Avalos for his valuable feedback during the preparation of this manuscript.

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

APPENDIX

# A  THEORETICAL DETAILS ON WAE-MDPS

## A.1  THE DISCREPANCY MEASURE

We show that reasoning about discrepancy measures between stationary distributions is sound in the context of infinite interaction and episodic RL processes. Let $P_\theta$ be a parameterized behavioral model that generate finite traces from the original environment (i.e., finite sequences of state, actions, and rewards of the form $\langle s_{0:T}, a_{0:T-1}, r_{0:T-1} \rangle$), our goal is to find the best parameter $\theta$ which offers the most accurate reconstruction of the original traces issued from the original model $\mathcal{M}$ operating under $\pi$. We demonstrate that, in the limit, considering the OT between trace-based distributions is equivalent to considering the OT between the stationary distribution of $\mathcal{M}_\pi$ and the one of the behavioral model.

Let us first formally recall the definition of the metric on the *transitions* of the MDP.

**Raw transition distance.** Assume that $\mathcal{S}$, $\mathcal{A}$, and $\mathrm{Im}(\mathcal{R})$ are respectively equipped with metric $d_\mathcal{S}$, $d_\mathcal{A}$, and $d_\mathcal{R}$, let us define the *raw transition distance metric* over *transitions* of $\mathcal{M}$, i.e., tuples of the form $\langle s, a, r, s' \rangle$, as $\vec{d} \colon \mathcal{S} \times \mathcal{A} \times \mathrm{Im}(\mathcal{R}) \times \mathcal{S}$,

$$\vec{d}\big(\langle s_1, a_1, r_1, s_1' \rangle, \langle s_2, a_2, r_2, s_2' \rangle\big) = d_\mathcal{S}(s_1, s_2) + d_\mathcal{A}(a_1, a_2) + d_\mathcal{R}(r_1, r_2) + d_\mathcal{S}\big(s_1', s_2'\big).$$

In a nutshell, $\vec{d}$ consists of the sum of the distance of all the transition components. Note that it is a well defined distance metric since the sum of distances preserves the identity of indiscernible, symmetry, and triangle inequality.

**Trace-based distributions.** The raw distance $\vec{d}$ allows to reason about *transitions*, we thus consider the distribution over *transitions which occur along traces of length $T$* to compare the dynamics of the original and behavioral models:

$$\mathcal{D}_\pi[T]\big(s, a, r, s'\big) = \frac{1}{T} \sum_{t=1}^{T} \xi_\pi^t(s \mid s_I) \cdot \pi(a \mid s) \cdot \mathbf{P}\big(s' \mid s, a\big) \cdot \mathbf{1}_{r = \mathcal{R}(s,a)}, \text{ and}$$

$$\mathcal{P}_\theta[T]\big(s, a, r, s'\big) = \frac{1}{T} \sum_{t=1}^{T} \mathop{\mathbb{E}}_{s_{0:t}, a_{0:t-1}, r_{0:t-1} \sim P_\theta[t]} \mathbf{1}_{\langle s_{t-1}, a_{t-1} r_{t-1}, s_t \rangle = \langle s, a, r, s' \rangle},$$

where $P_\theta[T]$ denotes the distribution over traces of length $T$, generated from $P_\theta$. Intuitively, $1/T \cdot \sum_{t=1}^{T} \xi_\pi^t(s \mid s_I)$ can be seen as the fraction of the time spent in $s$ along traces of length $T$, starting from the initial state Kulkarni (1995). Therefore, drawing $\langle s, a, r, s' \rangle \sim \mathcal{D}_\pi[T]$ trivially follows: it is equivalent to drawing $s$ from $1/T \cdot \sum_{t=1}^{T} \xi_\pi^t(\cdot \mid s_I)$, then respectively $a$ and $s'$ from $\pi(\cdot \mid s)$ and $\mathbf{P}(\cdot \mid s, a)$, to finally obtain $r = \mathcal{R}(s, a)$. Given $T \in \mathbb{N}$, our objective is to minimize the Wasserstein distance between those distributions: $W_{\vec{d}}(\mathcal{D}_\pi[T], \mathcal{P}_\theta[T])$. The following Lemma enables optimizing the Wasserstein distance between the original MDP and the behavioral model when traces are drawn from episodic RL processes or infinite interactions (Huang, 2020).

**Lemma A.1.** *Assume the existence of a stationary behavioral model $\xi_\theta = \lim_{T \to \infty} \mathcal{P}_\theta[T]$, then*

$$\lim_{T \to \infty} W_{\vec{d}}(\mathcal{D}_\pi[T], \mathcal{P}_\theta[T]) = W_{\vec{d}}(\xi_\pi, \xi_\theta).$$

*Proof.* First, note that $1/T \cdot \sum_{t=1}^{T} \xi_\pi^t(\cdot \mid s_I)$ weakly converges to $\xi_\pi$ as $T$ goes to $\infty$ Kulkarni (1995). The result follows then from (Villani, 2009, Corollary 6.9). $\square$

## A.2  DEALING WITH DISCRETE ACTIONS

When the policy $\pi$ executed in $\mathcal{M}$ already produces discrete actions, learning a latent action space is, in many cases, not necessary. We thus make the following assumptions:

**Assumption A.2.** *Let $\pi \colon \mathcal{S} \to \Delta(\mathcal{A}^\star)$ be the policy executed in $\mathcal{M}$ and assume that $\mathcal{A}^\star$ is a (tractable) finite set. Then, we take $\overline{\mathcal{A}} = \mathcal{A}^\star$ and $\phi_\iota^\mathcal{A}$ as the identity function, i.e., $\phi_\iota^\mathcal{A} \colon \overline{\mathcal{S}} \times \mathcal{A}^\star \to \mathcal{A}^\star, \langle \bar{s}, a^\star \rangle \mapsto a^\star$.*

**Assumption A.3.** *Assume that the action space of the original environment $\mathcal{M}$ is a (tractable) finite set. Then, we take $\psi_\theta$ as the identity function, i.e., $\psi_\theta = \phi_\iota^{\mathcal{A}}$.*

Concretely, the premise of Assumption A.2 typically occurs when $\pi$ is a latent policy (see Rem. 1) *or* when $\mathcal{M}$ has already a discrete action space. In the latter case, Assumption A.2 and A.3 amount to setting $\overline{\mathcal{A}} = \mathcal{A}$ and ignoring the action encoder and embedding function. Note that if a discrete action space is too large, or if the user explicitly aims for a coarser space, then the former is not considered as tractable, these assumptions do not hold, and the action space is abstracted to a smaller set of discrete actions.

## A.3  Proof of Lemma 3.2

**Notation.** From now on, we write $\phi_\iota(\bar{s}, \bar{a} \mid s, a) = \mathbf{1}_{\phi_\iota(s) = \bar{s}} \cdot \phi_\iota^{\mathcal{A}}(\bar{a} \mid \bar{s}, a)$.

**Lemma 3.2.** *Define $\mathcal{T}(\bar{s}, \bar{a}, \bar{s}') = \mathbb{E}_{s, a \sim \xi_\pi}[\mathbf{1}_{\phi_\iota(s) = \bar{s}} \cdot \phi_\iota^{\mathcal{A}}(\bar{a} \mid \bar{s}, a) \cdot \overline{\mathbf{P}}_\theta(\bar{s}' \mid \bar{s}, \bar{a})]$ as the distribution of drawing state-action pairs from interacting with $\mathcal{M}$, embedding them to the latent spaces, and finally letting them transition to their successor state in $\overline{\mathcal{M}}_\theta$. Then, $W_{\vec{d}}\left(Q_\iota, \bar{\xi}_{\overline{\pi}_\theta}\right) \leqslant W_{\vec{d}}\left(\bar{\xi}_{\overline{\pi}_\theta}, \mathcal{T}\right) + L_{\mathbf{P}}^{\xi_\pi}.$*

*Proof.* Wasserstein is compliant with the triangular inequality (Villani, 2009), which gives us:
$$W_{\vec{d}}\left(Q_\iota, \bar{\xi}_{\overline{\pi}_\theta}\right) \leqslant W_{\vec{d}}\left(Q_\iota, \mathcal{T}\right) + W_{d_{\overline{\mathcal{S}}}}\left(\mathcal{T}, \bar{\xi}_{\overline{\pi}_\theta}\right),$$
where

$$W_{\vec{d}}\left(\mathcal{T}, \bar{\xi}_{\overline{\pi}_\theta}\right) \qquad\qquad \text{(note that } W_{\vec{d}} \text{ is reflexive (Villani, 2009))}$$
$$= \sup_{f \in \mathcal{F}_{\vec{d}}} \mathbb{E}_{s, a \sim \xi_\pi} \mathbb{E}_{\bar{s}, \bar{a} \sim \phi_\iota(\cdot \mid s, a)} \mathbb{E}_{\bar{s}' \sim \overline{\mathbf{P}}_\theta(\cdot \mid \bar{s}, \bar{a})} f\left(\bar{s}, \bar{a}, \bar{s}'\right) - \mathbb{E}_{\bar{s} \sim \bar{\xi}_{\overline{\pi}_\theta}} \mathbb{E}_{\bar{a} \sim \overline{\pi}_\theta(\cdot \mid \bar{s})} \mathbb{E}_{\bar{s}' \sim \overline{\mathbf{P}}_\theta(\cdot \mid \bar{s}, \bar{a})} f\left(\bar{s}, \bar{a}, \bar{s}'\right), \text{ and}$$

$$W_{\vec{d}}(Q_\iota, \mathcal{T})$$
$$= \sup_{f \in \mathcal{F}_{\vec{d}}} \mathbb{E}_{s, a, s' \sim \xi_\pi} \mathbb{E}_{\bar{s}, \bar{a}, \bar{s}' \sim \phi_\iota(\cdot \mid s, a, s')} f\left(\bar{s}, \bar{a}, \bar{s}'\right) - \mathbb{E}_{s, a \sim \xi_\pi} \mathbb{E}_{\bar{s}, \bar{a} \sim \phi_\iota(\cdot \mid s, a)} \mathbb{E}_{\bar{s}' \sim \overline{\mathbf{P}}_\theta(\cdot \mid \bar{s}, \bar{a})} f\left(\bar{s}, \bar{a}, \bar{s}'\right) \qquad (3)$$
$$\leqslant \mathbb{E}_{s, a \sim \xi_\pi} \mathbb{E}_{\bar{s}, \bar{a} \sim \phi_\iota(\cdot \mid s, a)} \sup_{f \in \mathcal{F}_{\vec{d}}} \mathbb{E}_{s' \sim \mathbf{P}(\cdot \mid s, a)} f\left(\bar{s}, \bar{a}, \phi_\iota\left(s'\right)\right) - \mathbb{E}_{\bar{s}' \sim \overline{\mathbf{P}}_\theta(\cdot \mid \bar{s}, \bar{a})} f\left(\bar{s}, \bar{a}, \bar{s}'\right) \qquad (4)$$
$$= \mathbb{E}_{s, a \sim \xi_\pi} \mathbb{E}_{\bar{a} \sim \phi_\iota^{\mathcal{A}}(\cdot \mid \phi_\iota(s), a)} \sup_{f \in \mathcal{F}_{d_{\overline{\mathcal{S}}}}} \mathbb{E}_{\bar{s}' \sim \phi_\iota \mathbf{P}(\cdot \mid s, a)} f\left(\bar{s}'\right) - \mathbb{E}_{\bar{s}' \sim \overline{\mathbf{P}}_\theta(\cdot \mid \phi_\iota(s), \bar{a})} f\left(\bar{s}'\right) \qquad (5)$$
$$= \mathbb{E}_{s, a \sim \xi_\pi} \mathbb{E}_{\bar{a} \sim \phi_\iota^{\mathcal{A}}(\cdot \mid \phi_\iota(s), a)} W_{d_{\overline{\mathcal{S}}}}\left(\phi_\iota \mathbf{P}(\cdot \mid s, a), \overline{\mathbf{P}}_\theta(\cdot \mid \phi_\iota(s), \bar{a})\right).$$

We pass from Eq. 3 to Eq. 4 by the Jensen's inequality. To see how we pass from Eq. 4 to Eq. 5, notice that
$$\mathcal{F}_{\vec{d}} = \left\{ f \colon f\left(\bar{s}_1, \bar{a}_1, \bar{s}_1'\right) - f\left(\bar{s}_2, \bar{a}_2, \bar{s}_2'\right) \leqslant \vec{d}\left(\langle \bar{s}_1, \bar{a}_1, \bar{s}_1' \rangle, \langle \bar{s}_2, \bar{a}_2, \bar{s}_2' \rangle\right) \right\}$$
$$\mathcal{F}_{\vec{d}} = \left\{ f \colon f\left(\bar{s}_1, \bar{a}_1, \bar{s}_1'\right) - f\left(\bar{s}_2, \bar{a}_2, \bar{s}_2'\right) \leqslant d_{\overline{\mathcal{S}}}(\bar{s}_1, \bar{s}_2) + d_{\overline{\mathcal{A}}}(\bar{a}_1, \bar{a}_2) + d_{\overline{\mathcal{S}}}\left(\bar{s}_1', \bar{s}_2'\right) \right\}$$
Observe now that $\bar{s}$ and $\bar{a}$ are fixed in the supremum computation of Eq. 4: all functions $f$ considered and taken from $\mathcal{F}_{\vec{d}}$ are of the form $f(\bar{s}, \bar{a}, \cdot)$. It is thus sufficient to consider the supremum over functions from the following subset of $\mathcal{F}_{\vec{d}}$:
$$\left\{ f \colon f\left(\bar{s}, \bar{a}, \bar{s}_1'\right) - f\left(\bar{s}, \bar{a}, \bar{s}_2'\right) \leqslant d_{\overline{\mathcal{S}}}(\bar{s}, \bar{s}) + d_{\overline{\mathcal{A}}}(\bar{a}, \bar{a}) + d_{\overline{\mathcal{S}}}\left(\bar{s}_1', \bar{s}_2'\right) \right\}$$
$$\text{(for } \bar{s}, \bar{a} \text{ drawn from } \phi_\iota)$$
$$= \left\{ f \colon f\left(\bar{s}, \bar{a}, \bar{s}_1'\right) - f\left(\bar{s}, \bar{a}, \bar{s}_2'\right) \leqslant d_{\overline{\mathcal{S}}}\left(\bar{s}_1', \bar{s}_2'\right) \right\}$$
$$= \left\{ f \colon f\left(\bar{s}_1'\right) - f\left(\bar{s}_2'\right) \leqslant d_{\overline{\mathcal{S}}}\left(\bar{s}_1', \bar{s}_2'\right) \right\}$$
$$= \mathcal{F}_{d_{\overline{\mathcal{S}}}}.$$

Given a state $s \in \mathcal{S}$ in the original model, the (parallel) execution of $\pi$ in $\overline{\mathcal{M}}_\theta$ is enabled through $\pi(a, \bar{a} \mid s) = \pi(a \mid s) \cdot \phi_\iota^{\mathcal{A}}(\bar{a} \mid \phi_\iota(s), a)$ (cf. Fig. 1b). The local transition loss resulting from this interaction is:
$$L_{\mathbf{P}}^{\xi_\pi} = \mathbb{E}_{s, \langle a, \bar{a} \rangle \sim \xi_\pi} W_{d_{\overline{\mathcal{S}}}}\left(\phi_\iota \mathbf{P}(\cdot \mid s, a), \overline{\mathbf{P}}(\cdot \mid \phi_\iota(s), \bar{a})\right)$$
$$= \mathbb{E}_{s, a \sim \xi_\pi} \mathbb{E}_{\bar{a} \sim \phi_\iota^{\mathcal{A}}(\cdot \mid \phi_\iota(s), a)} W_{d_{\overline{\mathcal{S}}}}\left(\phi_\iota \mathbf{P}(\cdot \mid s, a), \overline{\mathbf{P}}_\theta(\cdot \mid \phi_\iota(s), \bar{a})\right),$$

which finally yields the result. $\qquad\square$

### A.4 PROOF OF THEOREM 3.3

Before proving Theorem 3.3, let us introduce the following Lemma, that explicitly demonstrates the link between the transition regularizer of the $\text{W}^2\text{AE-MDP}$ objective and the local transition loss required to obtain the guarantees related to the bisimulation bounds of Eq. 1.

**Lemma A.4.** *Assume that traces are generated by running $\bar{\pi} \in \overline{\Pi}$ in the original environment, then*

$$\mathbb{E}_{s,a^{\star} \sim \xi_{\bar{\pi}}} \mathbb{E}_{\bar{a} \sim \phi_{\iota}^{\mathcal{A}}(\cdot | \phi_{\iota}(s), a^{\star})} W_{d_{\overline{S}}}\left(\phi_{\iota}\mathbf{P}(\cdot \mid s, a^{\star}), \overline{\mathbf{P}}_{\theta}(\cdot \mid \phi_{\iota}(s), \bar{a})\right) = L_{\mathbf{P}}^{\xi_{\bar{\pi}}}.$$

*Proof.* Since the latent policy $\bar{\pi}$ generates latent actions, Assumption A.2 holds, which means:

$$\mathbb{E}_{s,a^{\star} \sim \xi_{\bar{\pi}}} \mathbb{E}_{\bar{a} \sim \phi_{\iota}^{\mathcal{A}}(\cdot | \phi_{\iota}(s), a^{\star})} W_{d_{\overline{S}}}\left(\phi_{\iota}\mathbf{P}(\cdot \mid s, a^{\star}), \overline{\mathbf{P}}_{\theta}(\cdot \mid \phi_{\iota}(s), \bar{a})\right)$$

$$= \mathbb{E}_{s,\bar{a} \sim \xi_{\bar{\pi}}} W_{d_{\overline{S}}}\left(\phi_{\iota}\mathbf{P}(\cdot \mid s, \bar{a}), \overline{\mathbf{P}}_{\theta}(\cdot \mid \phi_{\iota}(s), \bar{a})\right)$$

$$= L_{\mathbf{P}}^{\xi_{\bar{\pi}}}.$$

$\qquad\square$

**Theorem 3.3.** *Assume that traces are generated by running a latent policy $\bar{\pi} \in \overline{\Pi}$ in the original environment and let $d_{\mathcal{R}}$ be the usual Euclidean distance, then the $\text{W}^2\text{AE-MDP}$ objective is*

$$\min_{\iota,\theta} \mathbb{E}_{s,s' \sim \xi_{\bar{\pi}}} \left[ d_{\mathcal{S}}(s, \mathcal{G}_{\theta}(\phi_{\iota}(s))) + d_{\mathcal{S}}\left(s', \mathcal{G}_{\theta}\left(\phi_{\iota}(s')\right)\right) \right] + L_{\mathcal{R}}^{\xi_{\bar{\pi}}} + \beta \cdot (\mathcal{W}_{\xi_{\bar{\pi}}} + L_{\mathbf{P}}^{\xi_{\bar{\pi}}}).$$

*Proof.* We distinguish two cases: (i) the case where the original and latent models share the same discrete action space, i.e., $\mathcal{A} = \overline{\mathcal{A}}$, and (ii) the case where the two have a different action space (e.g., when the original action space is continuous), i.e., $\mathcal{A} \neq \overline{\mathcal{A}}$. In both cases, the local losses term follows by definition of $L_{\mathcal{R}}^{\xi_{\bar{\pi}}}$ and Lemma A.4. When $d_{\mathcal{R}}$ is the Euclidean distance (or even the $L_1$ distance since rewards are scalar values), the expected reward distance occurring in the expected trace-distance term $\vec{d}$ in the $\text{W}^2\text{AE-MDP}$ objective directly translates to the local loss $L_{\mathcal{R}}^{\xi_{\bar{\pi}}}$. Concerning the local transition loss, in case (i), the result naturally follows from Assumption A.2 and A.3. In case (ii), only Assumption A.2 holds, meaning the action encoder term of the $\text{W}^2\text{AE-}$ MDP objective is ignored, but not the action embedding term appearing in $G_{\theta}$. Given $s \sim \xi_{\bar{\pi}}$, recall that executing $\bar{\pi}$ in $\mathcal{M}$ amounts to embedding the produced latent actions $\bar{a} \sim \bar{\pi}(\cdot \mid \phi_{\iota}(s))$ back to the original environment via $a = \psi_{\theta}(\phi_{\iota}(s), \bar{a})$ (cf. Rem. 1 and Fig. 1a). Therefore, the projection of $\vec{d}(\langle s, a, r, s'\rangle, G_{\theta}(\phi_{\iota}(s), \bar{a}, \phi_{\iota}(s')))$ on the action space $\mathcal{A}$ is $d_{\mathcal{A}}(\psi_{\theta}(\phi_{\iota}(s), \bar{a}), \psi_{\theta}(\phi_{\iota}(s), \bar{a})) = 0$, for $r = \mathcal{R}(s, a)$ and $s' \sim \mathbf{P}(\cdot \mid s, a)$. $\qquad\square$

### A.5 OPTIMIZING THE TRANSITION REGULARIZER

In the following, we detail how we derive a tractable form of our transition regularizer $L_{\mathbf{P}}^{\xi_{\pi}}(\omega)$. Optimizing the ground Kantorovich-Rubinstein duality is enabled via the introduction of a parameterized, 1-Lipschitz network $\varphi_{\omega}^{\mathbf{P}}$, that need to be trained to attain the supremum of the dual:

$$L_{\mathbf{P}}^{\xi_{\pi}}(\omega) = \mathbb{E}_{s,a \sim \xi_{\pi}} \mathbb{E}_{\bar{s},\bar{a} \sim \phi_{\iota}(\cdot|s,a)} \max_{\omega: \varphi_{\omega}^{\mathbf{P}} \in \mathcal{F}_{d_{\overline{S}}}} \mathbb{E}_{\bar{s}' \sim \phi_{\iota}\mathbf{P}(\cdot|s,a)} \varphi_{\omega}^{\mathbf{P}}\left(\bar{s}'\right) - \mathbb{E}_{\bar{s}' \sim \overline{\mathbf{P}}_{\theta}(\cdot|\bar{s},\bar{a})} \varphi_{\omega}^{\mathbf{P}}\left(\bar{s}'\right).$$

Under this form, optimizing $L_{\mathbf{P}}^{\xi_{\pi}}(\omega)$ is intractable due to the expectation over the maximum. The following Lemma allows us rewriting $L_{\mathbf{P}}^{\xi_{\pi}}$ to make the optimization tractable through Monte Carlo estimation.

**Lemma A.5.** *Let $\mathcal{X}, \mathcal{Y}$ be two measurable sets, $\xi \in \Delta(\mathcal{X})$, $P: \mathcal{X} \to \Delta(\mathcal{Y}), Q: \mathcal{X} \to \Delta(\mathcal{Y})$, and $d: \mathcal{Y} \times \mathcal{Y} \to [0, +\infty[$ be a metric on $\mathcal{Y}$. Then,*

$$\mathbb{E}_{x \sim \xi} W_d\left(P(\cdot \mid x), Q(\cdot \mid x)\right) = \sup_{\varphi: \mathcal{X} \to \mathcal{F}_d} \mathbb{E}_{x \sim \xi} \left[ \mathbb{E}_{y_1 \sim P(\cdot|x)} \varphi(x)(y_1) - \mathbb{E}_{y_2 \sim Q(\cdot|x)} \varphi(x)(y_2) \right]$$

*Proof.* Our objective is to show that

$$
\mathbb{E}_{x \sim \xi} \left[ \sup_{f \in \mathcal{F}_d} \mathbb{E}_{y_1 \sim P(\cdot|x)} \varphi(y_1)(x) - \mathbb{E}_{y_2 \sim Q(\cdot|x)} \varphi(y_2)(x) \right] \tag{6}
$$

$$
= \sup_{\varphi \colon \mathcal{X} \to \mathcal{F}_d} \mathbb{E}_{x \sim \xi} \left[ \mathbb{E}_{y_1 \sim P(\cdot|x)} \varphi(x)(y_1) - \mathbb{E}_{y_2 \sim Q(\cdot|x)} \varphi(x)(y_2) \right] \tag{7}
$$

We start with (6) $\leqslant$ (7). Construct $\varphi^\star \colon \mathcal{X} \to \mathcal{F}_d$ by setting for all $x \in \mathcal{X}$

$$
\varphi^\star(x) = \arg \sup_{f \in \mathcal{F}_d} \mathbb{E}_{y_1 \sim P(\cdot|x)} f(y_1) - \mathbb{E}_{y_2 \sim Q(\cdot|x)} f(y_2).
$$

This gives us

$$
\mathbb{E}_{x \sim \xi} \left[ \sup_{f \in \mathcal{F}_d} \mathbb{E}_{y_1 \sim P(\cdot|x)} f(y_1) - \mathbb{E}_{y_2 \sim Q(\cdot|x)} f(y_2) \right]
$$

$$
= \mathbb{E}_{x \sim \xi} \left[ \mathbb{E}_{y_1 \sim P(\cdot|x)} \varphi^\star(x)(y_1) - \mathbb{E}_{y_2 \sim Q(\cdot|x)} \varphi^\star(x)(y_2) \right]
$$

$$
\leqslant \sup_{\varphi \colon \mathcal{X} \to \mathcal{F}_d} \mathbb{E}_{x \sim \xi} \left[ \mathbb{E}_{y_1 \sim P(\cdot|x)} \varphi(x)(y_1) - \mathbb{E}_{y_2 \sim Q(\cdot|x)} \varphi(x)(y_2) \right].
$$

It remains to show that (6) $\geqslant$ (7). Take

$$
\varphi^\star = \arg \sup_{\varphi \colon \mathcal{X} \to \mathcal{F}_d} \mathbb{E}_{x \sim \xi} \left[ \mathbb{E}_{y_1 \sim P(\cdot|x)} \varphi(x)(y_1) - \mathbb{E}_{y_2 \sim Q(\cdot|x)} \varphi(x)(y_2) \right].
$$

Then, for all $x \in \mathcal{X}$, we have $\varphi^\star(x) \in \mathcal{F}_d$ which means:

$$
\mathbb{E}_{y_1 \sim P(\cdot|x)} \varphi^\star(x)(y_1) - \mathbb{E}_{y_2 \sim Q(\cdot|x)} \varphi^\star(x)(y_2)
$$

$$
\leqslant \sup_{f \in \mathcal{F}_d} \mathbb{E}_{y_1 \sim P(\cdot|x)} f(y_1) - \mathbb{E}_{y_2 \sim Q(\cdot|x)} f(y_2)
$$

This finally yields

$$
\mathbb{E}_{x \sim \xi} \left[ \mathbb{E}_{y_1 \sim P(\cdot|x)} \varphi^\star(x)(y_1) - \mathbb{E}_{y_2 \sim Q(\cdot|x)} \varphi^\star(x)(y_2) \right]
$$

$$
\leqslant \mathbb{E}_{x \sim \xi} \left[ \sup_{f \in \mathcal{F}_d} \mathbb{E}_{y_1 \sim P(\cdot|x)} f(y_1) - \mathbb{E}_{y_2 \sim Q(\cdot|x)} f(y_2) \right].
$$

$\square$

**Corollary A.5.1.** *Let $\xi_\pi$ be a stationary distribution of $\mathcal{M}_\pi$ and $\mathcal{X} = \mathcal{S} \times \mathcal{A} \times \bar{\mathcal{S}} \times \bar{\mathcal{A}}$, then*

$$
L_{\mathbf{P}}^{\xi_\pi} = \sup_{\varphi \colon \mathcal{X} \to \mathcal{F}_{d_{\bar{\mathcal{S}}}}} \mathbb{E}_{s,a,s' \sim \xi_\pi} \mathbb{E}_{\bar{s},\bar{a} \sim \phi_\iota(\cdot|s,a)} \left[ \varphi(s,a,\bar{s},\bar{a})\big(\phi_\iota(s')\big) - \mathbb{E}_{\bar{s}' \sim \bar{\mathbf{P}}_\theta(\cdot|\bar{s},a)} \varphi(s,a,\bar{s},\bar{a})\big(\bar{s}'\big) \right]
$$

Consequently, we rewrite $L_{\mathbf{P}}^{\xi_\pi}(\omega)$ as a tractable maximization:

$$
L_{\mathbf{P}}^{\xi_\pi}(\omega) = \max_{\omega \colon \varphi_\omega^{\mathbf{P}} \in \mathcal{F}_{d_{\bar{\mathcal{S}}}}} \mathbb{E}_{s,a,s' \sim \xi_\pi} \mathbb{E}_{\bar{s},\bar{a} \sim \phi_\iota(\cdot|s,a)} \left[ \varphi_\omega^{\mathbf{P}}\big(s,a,\bar{s},\bar{a},\phi_\iota(s')\big) - \mathbb{E}_{\bar{s}' \sim \bar{\mathbf{P}}_\theta(\cdot|\bar{s},\bar{a})} \varphi_\omega^{\mathbf{P}}\big(s,a,\bar{s},\bar{a},\bar{s}'\big) \right].
$$

### A.6 THE LATENT METRIC

In the following, we show that considering the Euclidean distance for $\vec{d}$ and $d_{\bar{\mathcal{S}}}$ in the latent space for optimizing the regularizers $\mathcal{W}_{\xi_\pi}$ and $L_{\mathbf{P}}^{\xi_\pi}$ is Lipschitz equivalent to considering a continuous $\lambda$-relaxation of the *discrete metric* $\mathbf{1}_{\neq}(\boldsymbol{x},\boldsymbol{y}) = \mathbf{1}_{\boldsymbol{x} \neq \boldsymbol{y}}$. Consequently, this also means it is consistently sufficient to enforce 1-Lispchitzness via the gradient penalty approach of Gulrajani et al. (2017) during training to maintain the guarantees linked to the regularizers in the zero-temperature limit, when the spaces are discrete.

**Lemma A.6.** *Let $d$ be the usual Euclidean distance and $d_\lambda \colon [0,1]^n \times [0,1]^n \to [0,1[, \langle x, y \rangle \mapsto \frac{d(x,y)}{\lambda + d(x,y)}$ for $\lambda \in ]0,1]$ and $n \in \mathbb{N}$, then $d_\lambda$ is a distance metric.*

*Proof.* The function $d_\lambda$ is a metric iff it satisfies the following axioms:

1. *Identity of indiscernibles*: If $x = y$, then $d_\lambda(x, y) = \frac{d(x,y)}{\lambda + d(x,y)} = \frac{0}{\lambda + 0} = 0$ since $d$ is a distance metric. Assume now that $d_\lambda(x, y) = 0$ and take $\alpha = d(x, y)$, for any $x, y$. Thus, $\alpha \in [0, +\infty[$ and $0 = \frac{\alpha}{\lambda + \alpha}$ is only achieved in $\alpha = 0$, which only occurs whenever $x = y$ since $d$ is a distance metric.

2. *Symmetry*:

$$
\begin{aligned}
d_\lambda(x, y) &= \frac{d(x, y)}{\lambda + d(x, y)} \\
&= \frac{d(y, x)}{\lambda + d(y, x)} \qquad (d \text{ is a distance metric}) \\
&= d_\lambda(y, x)
\end{aligned}
$$

3. *Triangle inequality*: Let $x, y, z \in [0,1]^n$, the triangle inequality holds iff

$$d_\lambda(x, y) + d_\lambda(y, z) \geqslant d_\lambda(x, z) \tag{8}$$

$$\equiv \qquad \frac{d(x, y)}{\lambda + d(x, y)} + \frac{d(y, z)}{\lambda + d(y, z)} \geqslant \frac{d(x, z)}{\lambda + d(x, z)}$$

$$\equiv \qquad \frac{\lambda d(x, y) + \lambda d(y, z) + 2 d(x, y) d(y, z)}{\lambda^2 + \lambda d(x, y) + \lambda d(y, z) + d(x, y) d(y, z)} \geqslant \frac{d(x, z)}{\lambda + d(x, z)}$$

$$
\begin{aligned}
\equiv \quad & \lambda^2 d(x, y) + \lambda^2 d(y, z) + 2\lambda d(x, y) d(y, z) + \\
& \lambda d(x, y) d(x, z) + \lambda d(y, z) d(x, z) + 2 d(x, y) d(y, z) d(x, z) \\
\geqslant & \lambda^2 d(x, z) + \lambda d(x, y) d(x, z) + \lambda d(y, z) d(x, z) + d(x, y) d(y, z) d(x, z) \\
& \qquad (\text{cross-product, with } \lambda > 0 \text{ and } \operatorname{Im}(d) \in [0, \infty[)
\end{aligned}
$$

$$\equiv \quad \lambda^2 d(x, y) + \lambda^2 d(y, z) + 2\lambda d(x, y) d(y, z) + d(x, y) d(y, z) d(x, z) \geqslant \lambda^2 d(x, z) \tag{9}$$

Since $d$ is a distance metric, we have

$$\lambda^2 d(x, y) + \lambda^2 d(y, z) \geqslant \lambda^2 d(x, z) \tag{10}$$

and $\operatorname{Im}(d) \in [0, \infty[$, meaning

$$2\lambda d(x, y) d(y, z) + d(x, y) d(y, z) d(x, z) \geqslant 0 \tag{11}$$

By Eq. 10 and 11, the inequality of Eq. 9 holds. Furthermore, the fact that Eq. 8 and 9 are equivalent yields the result. $\qquad \square$

**Lemma A.7.** *Let $d$, $d_\lambda$ as defined above, then (i) $d_\lambda \xrightarrow[\lambda \to 0]{} \mathbf{1}_{\neq}$ and (ii) $d, d_\lambda$ are Lipschitz-equivalent.*

*Proof.* Part (i) is straightforward by definition of $d_\lambda$. Distances $d$ and $d_\lambda$ are Lispchitz equivalent if and only if $\exists a, b > 0$ such that $\forall x, y \in [0,1]^n$,

$$
\begin{aligned}
a \cdot d(x, y) \leqslant \; & d_\lambda(x, y) && \leqslant b \cdot d(x, y) \\
\equiv a \cdot d(x, y) \leqslant \; & \frac{d(x, y)}{\lambda + d(x, y)} && \leqslant b \cdot d(x, y) \\
\equiv \qquad a \leqslant \; & \frac{1}{\lambda + d(x, y)} && \leqslant b
\end{aligned}
$$

Taking $a = \frac{1}{\lambda + \sqrt{n}}$ and $b = \frac{1}{\lambda}$ yields the result. $\qquad \square$

**Corollary A.7.1.** *For all $\beta \geqslant 1/\lambda$, $s \in \mathcal{S}$, $a \in \mathcal{A}$, $\bar{s} \in \bar{\mathcal{S}}$, and $\bar{a} \in \bar{\mathcal{A}}$, we have*

1. $W_{d_\lambda}\left(\mathcal{T}, \bar{\xi}_{\bar{\pi}_\theta}\right) \leqslant \beta \cdot W_d\left(\mathcal{T}, \bar{\xi}_{\bar{\pi}_\theta}\right)$

2. $W_{d_\lambda}\left(\phi_\iota \mathbf{P}(\cdot \mid s, a), \overline{\mathbf{P}}_\theta(\cdot \mid \bar{s}, \bar{a})\right) \leqslant \beta \cdot W_d\left(\phi_\iota \mathbf{P}(\cdot \mid s, a), \overline{\mathbf{P}}_\theta(\cdot \mid \bar{s}, \bar{a})\right)$

*Proof.* By Lipschitz equivalence, taking $\beta \geqslant 1/\lambda$ ensures that $\forall n \in \mathbb{N}$, $\forall \boldsymbol{x}, \boldsymbol{y} \in [0,1]^n$, $d_\lambda(\boldsymbol{x}, \boldsymbol{y}) \leqslant \beta \cdot d(\boldsymbol{x}, \boldsymbol{y})$. Moreover, for any distributions $P, Q$, $W_{d_\lambda}(P, Q) \leqslant \beta \cdot W_d(P, Q)$ (cf., e.g., Gelada et al. 2019, Lemma A.4 for details). $\qquad\square$

In practice, taking the hyperparameter $\beta \geqslant 1/\lambda$ in the $\text{W}^2\text{AE-MDP}$ ensures that minimizing the $\beta$-scaled regularizers w.r.t. $d$ also minimizes the regularizers w.r.t. the $\lambda$-relaxation $d_\lambda$, being the discrete distribution in the zero-temperature limit. Note that optimizing over two different $\beta_1, \beta_2$ instead of a unique scale factor $\beta$ is also a good practice to interpolate between the two regularizers.

## B  EXPERIMENT DETAILS

The code for conducting and replicating our experiments is available at `https://github.com/florentdelgrange/wae_mdp`.

### B.1  SETUP

We used TENSORFLOW `2.7.0` (Abadi et al., 2015) to implement the neural network architecture of our $\text{W}^2\text{AE-MDP}$, TENSORFLOW PROBABILITY `0.15.0` (Dillon et al., 2017) to handle the probabilistic components of the latent model (e.g., latent distributions with reparameterization tricks, masked autoregressive flows, etc.), as well as TF-AGENTS `0.11.0` (Guadarrama et al., 2018) to handle the RL parts of the framework.

Models have been trained on a cluster running under `CentOS Linux 7 (Core)` composed of a mix of nodes containing Intel processors with the following CPU microarchitectures: (i) `10-core INTEL E5-2680v2`, (ii) `14-core INTEL E5-2680v4`, and (iii) `20-core INTEL Xeon Gold 6148`. We used 8 cores and 32 GB of memory for each run.

### B.2  STATIONARY DISTRIBUTION

To sample from the stationary distribution $\xi_\pi$ of episodic learning environments operating under $\pi \in \Pi$, we implemented the *recursive $\epsilon$-perturbation trick* of Huang (2020). In a nutshell, the reset of the environment is explicitly added to the state space of $\mathcal{M}$, which is entered at the end of each episode and left with probability $1 - \epsilon$ to start a new one. We also added a special atomic proposition reset into $\mathbf{AP}$ to label this reset state and reason about episodic behaviors. For instance, this allows verifying whether the agent behaves safely during the entire episode, or if it is able to reach a goal before the end of the episode.

### B.3  ENVIRONMENTS WITH INITIAL DISTRIBUTION

Many environments do not necessarily have a single initial state, but rather an initial distribution over states $d_I \in \Delta(\mathcal{S})$. In that case, the results presented in this paper remain unchanged: it suffices to add a dummy state $s^\star$ to the state space $\mathcal{S} \cup \{s^\star\}$ so that $s_I = s^\star$ with the transition dynamics $\mathbf{P}(s' \mid s^\star, a) = d_I(s')$ for any action $a \in \mathcal{A}$. Therefore, each time the reset of the environment is triggered, we make the MDP entering the initial state $s^\star$, then transitioning to $s'$ according to $d_I$.

### B.4  LATENT SPACE DISTRIBUTION

As pointed out in Sect. 4, posterior collapse is naturally avoided when optimizing $\text{W}^2\text{AE-MDP}$. To illustrate that, we report the distribution of latent states produced by $\phi_\iota$ during training (Fig. 5). The plots reveal that the latent space generated by mapping original states drawn from $\xi_\pi$ during training to $\bar{\mathcal{S}}$ via $\phi_\iota$ is fairly distributed, for each environment.

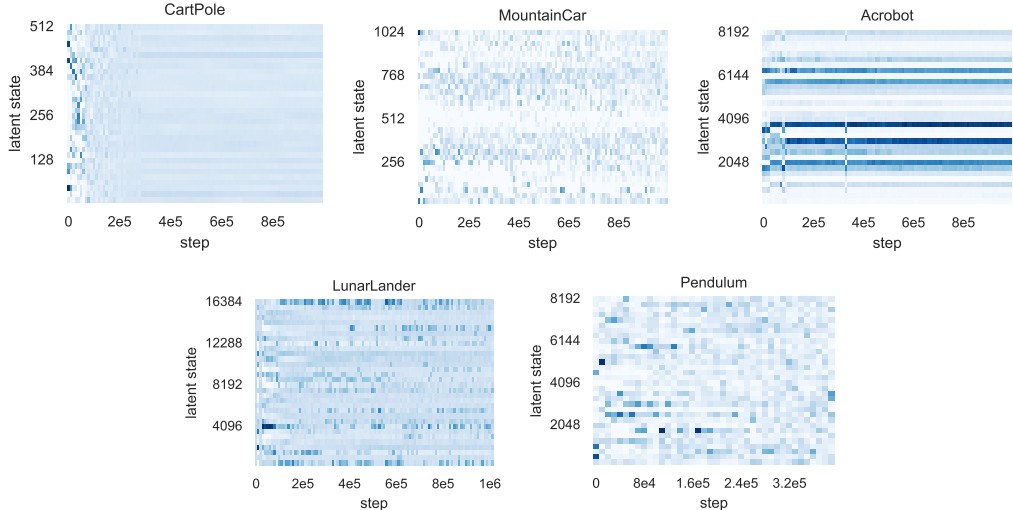

Figure 5: Latent space distribution along training steps. The intensity of the blue hue corresponds to the frequency of latent states produced by $\phi_\iota$ during training.

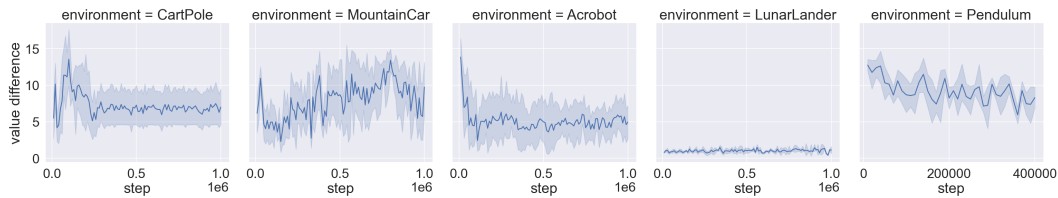

Figure 6: Absolute value difference $\|V_{\overline{\pi}_\theta}\|$ reported along training steps.

### B.5 DISTANCE METRICS: STATE, ACTION, AND REWARD RECONSTRUCTION

The choice of the distance functions $d_{\mathcal{S}}$, $d_{\mathcal{A}}$, and $d_{\mathcal{R}}$, plays a role in the success of our approach. The usual Euclidean distance is often a good choice for all the transition components, but the scale, dimensionality, and nature of the inputs sometimes require using scaled, normalized, or other kinds of distances to allow the network to reconstruct each component. While we did not observe such requirements in our experiments (where we simply used the Euclidean distance), high dimensional observations (e.g., images) are an example of data which could require tuning the state-distance function in such a way, to make sure that the optimization of the reward or action reconstruction will not be disfavored compared to that of the states.

### B.6 VALUE DIFFERENCE

In addition to reporting the quality guarantees of the model along training steps through local losses (cf. Figure 4b), our experiments revealed that the absolute value difference $\|V_{\overline{\pi}_\theta}\|$ between the original and latent models operating under the latent policy quickly decreases and tends to converge to values in the same range (Figure 6). This is consistent with the fact that minimizing local losses lead to close behaviors (cf. Eq. 1) and that the value function is Lipschitz-continuous w.r.t. $\widetilde{d}_{\overline{\pi}_\theta}$ (cf. Section 2).

### B.7 REMARK ON FORMAL VERIFICATION

Recall that *our bisimulation guarantees come by construction of the latent space*. Essentially, our learning algorithm spits out a distilled policy and a latent state space which already yields a guaranteed bisimulation distance between the original MDP and the latent MDP. This is the crux of how we enable verification techniques like model checking. In particular, bisimulation guarantees mean that *reachability probabilities in the latent MDP compared to those in the original one are close*.

Furthermore, the value difference of (omega-regular) properties (formulated through mu-calculus) obtained in the two models is bounded by this distance (cf. Sect. 2 and Chatterjee et al. 2010).

**Reachability is the key ingredient** to model-check MDPs. Model-checking properties is in most cases performed by reduction to the reachability of components or regions of the MDP: it either consists of (i) iteratively checking the reachability of the parts of the state space satisfying path formulae that comprise the specification, through a tree-like decomposition of the latter (e.g., for (P,R-)CTL properties, cf. Baier & Katoen 2008), or (ii) checking the reachability to the part of the state space of a product of the MDP with a memory structure or an automaton that embeds the omega-regular property — e.g., for LTL (Baier et al., 2016; Sickert et al., 2016), LTLf (Wells et al., 2020), or GLTL (Littman et al., 2017), among other specification formalisms. The choice of specification formalism is up to the user and depends on the case study. The scope of this work is focusing on learning to distill RL policies with bisimulation guarantees *so that model checking can be applied*, in order to reason about the behaviors of the agent. That being said, *reachability is all we need* to show that model checking can be applied.

### B.8 HYPERPARAMETERS

**W$^2$AE-MDP parameters.** All components (e.g., functions or distribution locations and scales, see Fig. 2) are represented and inferred by neural networks (multilayer perceptrons). All the networks share the same architecture (i.e., number of layers and neurons per layer). We use a simple uniform experience replay of size $10^6$ to store the transitions and sample them. The training starts when the agent has collected $10^4$ transitions in $\mathcal{M}$. We used minibatches of size $128$ to optimize the objective and we applied a minibatch update every time the agent executing $\pi$ has performed $16$ steps in $\mathcal{M}$. We use the recursive $\epsilon$-perturbation trick of Huang (2020) with $\epsilon = 3/4$: when an episode ends, it restarts from the initial state with probability $1/4$; before re-starting an episode, the time spent in the reset state labeled with reset follows then the geometric distribution with expectation $\epsilon/1-\epsilon = 3$. We chose the same latent state-action space size than Delgrange et al. (2022), except for LunarLander that we decreased to $\log_2 |\overline{\mathcal{S}}| = 14$ and $|\overline{\mathcal{A}}| = 3$ to improve the scalability of the verification.

**VAE-MDPs parameters.** For the comparison of Sect. 4, we used the exact same VAE-MDP hyperparameter set as prescribed by Delgrange et al. (2022), except for the state-action space of LunarLander that we also changed for scalability and fair comparison purpose.[2]

**Hyperparameter search.** To evaluate our W$^2$AE-MDP, we realized a search in the parameter space defined in Table 2. The best parameters found (in terms of trade-off between performance and latent quality) are reported in Table 3. We used two different optimizers for minimizing the loss (referred to as the minimizer) and computing the Wasserstein terms (reffered to as the maximizer). We used ADAM (Kingma & Ba, 2015) for the two, but we allow for different learning rates ADAM$_\alpha$ and exponential decays ADAM$_{\beta_1}$, ADAM$_{\beta_2}$. We also found that polynomial decay for ADAM$_\alpha$ (e.g., to $10^{-5}$ for $4 \cdot 10^5$ steps) is a good practice to stabilize the experiment learning curves, but is not necessary to obtain high-quality and performing distillation. Concerning the continuous relaxation of discrete distributions, we used a different temperature for each distribution, as Maddison et al. (2017) pointed out that doing so is valuable to improve the results. We further followed the guidelines of Maddison et al. (2017) to choose the interval of temperatures and did not schedule any annealing scheme (in contrast to VAE-MDPs). Essentially, the search reveals that the regularizer scale factors $\beta_\cdot$ (defining the optimization direction) as well as the encoder and latent transition temperatures are important to improve the performance of distilled policies. For the encoder temperature, we found a nice spot in $\lambda_{\phi_\iota} = 2/3$, which provides the best performance in general, whereas the choice of $\lambda_{\overline{P}_\theta}$ and $\beta_\cdot$ are (latent-) environment dependent. The importance of the temperature parameters for the continuous relaxation of discrete distributions is consistent with the results of (Maddison et al., 2017), revealing that the success of the relaxation depends on the choice of the temperature for the different latent space sizes.

**Labeling functions.** We used the same labeling functions as those described by Delgrange et al. (2022). For completeness, we recall the labeling function used for each environment in Table 4.

---

[2]The code for conducting the VAE-MDPs experiments is available at `https://github.com/florentdelgrange/vae_mdp` (GNU General Public License v3.0).

Table 2: Hyperparameter search. $\lambda_X$ refers to the temperature used for $\text{W}^2\text{AE-MDP}$ component $X$.

| Parameter | Range |
|---|---|
| $\text{ADAM}_\alpha$ (minimizer) | $\{\,0.0001, 0.0002, 0.0003, 0.001\,\}$ |
| $\text{ADAM}_\alpha$ (maximizer) | $\{\,0.0001, 0.0002, 0.0003, 0.001\,\}$ |
| $\text{ADAM}_{\beta_1}$ | $\{\,0, 0.5, 0.9\,\}$ |
| $\text{ADAM}_{\beta_2}$ | $\{\,0.9, 0.999\,\}$ |
| neurons per layer | $\{\,64, 128, 256, 512\,\}$ |
| number of hidden layers | $\{\,1, 2, 3\,\}$ |
| activation | $\{\,\text{ReLU}, \text{Leaky ReLU}, \tanh, \frac{\text{softplus}(2x+2)}{2} - 1\,(\textit{smooth ELU})\,\}$ |
| $\beta_{\mathcal{W}_{\xi_\pi}}$ | $\{\,10, 25, 50, 75, 100\,\}$ |
| $\beta_{L_{\mathbf{P}}^{\xi_\pi}}$ | $\{\,10, 25, 50, 75, 100\,\}$ |
| $m$ | $\{\,5, 10, 15, 20\,\}$ |
| $\delta$ | $\{\,10, 20\,\}$ |
| use $\varepsilon$-mimic (cf. Delgrange et al. 2022) | $\{\,\text{True}, \text{False}\,\}$ (if True, a decay rate of $10^{-5}$ is used) |
| $\lambda_{\overline{\mathbf{P}}_\theta}$ | $\{\,0.1, 1/3, 1/2, 2/3, 3/5, 0.99\,\}$ |
| $\lambda_{\phi_\iota}$ | $\{\,0.1, 1/3, 1/2, 2/3, 3/5, 0.99\,\}$ |
| $\lambda_{\overline{\pi}_\theta}$ | $\{\,1/|\overline{\mathcal{A}}|-1, 1/(|\overline{\mathcal{A}}|-1)\cdot 1.5\,\}$ |
| $\lambda_{\phi_\iota^{\mathcal{A}}}$ | $\{\,1/|\overline{\mathcal{A}}|-1, 1/(|\overline{\mathcal{A}}|-1)\cdot 1.5\,\}$ |

Table 3: Final hyperparameters used to evaluate $\text{W}^2\text{AE-MDPs}$ in Sect. 4

| | CartPole | MountainCar | Acrobot | LunarLander | Pendulum |
|---|---|---|---|---|---|
| $\log_2 |\overline{\mathcal{S}}|$ | 9 | 10 | 13 | 14 | 13 |
| $|\overline{\mathcal{A}}|$ | $2 = |\mathcal{A}|$ | $2 = |\mathcal{A}|$ | $3 = |\mathcal{A}|$ | 3 | 3 |
| activation | tanh | ReLU | Leaky Relu | ReLU | ReLU |
| layers | $[64, 64, 64]$ | $[512, 512]$ | $[512, 512]$ | $[256]$ | $[256, 256, 256]$ |
| $\text{ADAM}_\alpha$ (minimizer) | 0.0002 | 0.0001 | 0.0002 | 0.0003 | 0.0003 |
| $\text{ADAM}_\alpha$ (maximizer) | 0.0002 | 0.0001 | 0.0001 | 0.0003 | 0.0003 |
| $\text{ADAM}_{\beta_1}$ | 0.5 | 0 | 0 | 0 | 0.5 |
| $\text{ADAM}_{\beta_2}$ | 0.999 | 0.999 | 0.999 | 0.999 | 0.999 |
| $\beta_{L_{\mathbf{P}}^{\xi_\pi}}$ | 10 | 25 | 10 | 50 | 25 |
| $\beta_{\mathcal{W}_{\xi_\pi}}$ | 75 | 100 | 10 | 100 | 25 |
| $m$ | 5 | 20 | 20 | 15 | 5 |
| $\delta$ | 20 | 10 | 20 | 20 | 10 |
| $\varepsilon$ | 0 | 0 | 0 | 0 | 0.5 |
| $\lambda_{\overline{\mathbf{P}}_\theta}$ | $1/3$ | $1/3$ | 0.1 | 0.75 | $2/3$ |
| $\lambda_{\phi_\iota}$ | $1/3$ | $2/3$ | $2/3$ | $2/3$ | $2/3$ |
| $\lambda_{\overline{\pi}_\theta}$ | $2/3$ | $1/3$ | 0.5 | 0.5 | 0.5 |
| $\lambda_{\phi_\iota^{\mathcal{A}}}$ | / | / | / | $1/3$ | $1/3$ |

| Environment | $\mathcal{S} \subseteq$ | Description, for $s \in \mathcal{S}$ | $\ell(s) = \langle p_1, \dots, p_n, p_{\text{reset}} \rangle$ |
|---|---|---|---|
| CartPole | $\mathbb{R}^4$ | • $s_1$: cart position
• $s_2$: cart velocity
• $s_3$: pole angle (rad)
• $s_4$: pole velocity at tip | • $p_1 = \mathbf{1}_{s_1 \geqslant 1.5}$: unsafe cart position
• $p_2 = \mathbf{1}_{s_3 \geqslant 0.15}$: unsafe pole angle |
| MountainCar | $\mathbb{R}^2$ | • $s_1$: position
• $s_2$: velocity | • $p_1 = \mathbf{1}_{s_1 > 1.5}$: target position
• $p_2 = \mathbf{1}_{s_1 \geqslant -1/2}$: right-hand side of the mountain
• $p_3 = \mathbf{1}_{s_2 \geqslant 0}$: car going forward |
| Acrobot | $\mathbb{R}^6$ | Let $\theta_1, \theta_2 \in [0, 2\pi]$ be the angles of the two rotational joints,
• $s_1 = \cos(\theta_1)$
• $s_2 = \sin(\theta_1)$
• $s_3 = \cos(\theta_2)$
• $s_4 = \sin(\theta_2)$
• $s_5$: angular velocity 1
• $s_6$: angular velocity 2 | • $p_1 = \mathbf{1}_{-s_1 - s_3 \cdot s_1 + s_4 \cdot s_2 > 1}$: RL agent target
• $p_2 = \mathbf{1}_{s_1 \geqslant 0}$: $\theta_1 \in [0, \pi/2] \cup [3\pi/2, 2\pi]$
• $p_3 = \mathbf{1}_{s_2 \geqslant 0}$: $\theta_1 \in [0, \pi]$
• $p_4 = \mathbf{1}_{s_3 \geqslant 0}$: $\theta_2 \in [0, \pi/2] \cup [3\pi/2, 2\pi]$
• $p_5 = \mathbf{1}_{s_4 \geqslant 0}$: $\theta_2 \in [0, \pi]$
• $p_6 = \mathbf{1}_{s_5 \geqslant 0}$: positive angular velocity (1)
• $p_7 = \mathbf{1}_{s_6 \geqslant 0}$: positive angular velocity (2) |
| Pendulum | $\mathbb{R}^3$ | Let $\theta \in [0, 2\pi]$ be the joint angle
• $s_1 = \cos(\theta)$
• $s_2 = \sin(\theta)$
• $s_3$: angular velocity | • $p_1 = \mathbf{1}_{s_1 \geqslant \cos(\pi/3)}$: safe joint angle
• $p_2 = \mathbf{1}_{s_1 \geqslant 0}$: $\theta \in [0, \pi/2] \cup [3\pi/2, 2\pi]$
• $p_3 = \mathbf{1}_{s_2 \geqslant 0}$: $\theta \in [0, \pi]$
• $p_4 = \mathbf{1}_{s_3 \geqslant 0}$: positive angular velocity |
| LunarLander | $\mathbb{R}^8$ | • $s_1$: horizontal coordinates
• $s_2$: vertical coordinates
• $s_3$: horizontal speed
• $s_4$: vertical speed
• $s_5$: ship angle
• $s_6$: angular speed
• $s_7$: left leg contact
• $s_8$: right leg contact | • $p_1$: unsafe angle
• $p_2$: leg ground contact
• $p_3$: lands rapidly
• $p_4$: left inclination
• $p_5$: right inclination
• $p_6$: motors shut down |

Table 4: Labeling functions for the OpenAI environments considered in our experiments (Delgrange et al., 2022). We provide a short description of the state space and the meaning of each atomic proposition. Recall that labels are binary encoded, for $n = |\mathbf{AP}| - 1$ (one bit is reserved for reset) and $p_{\text{reset}} = 1$ iff $s$ is a reset state (cf. Appendix B.2).

**Time to failure properties.** Based on the labeling described in Table 4, we formally detail the time to failure properties checked in Sect. 4 whose results are listed in Table 1 for each environment. Let Reset $= \{\text{reset}\} = \langle 0, \dots, 1 \rangle$ (we assume here that the last bit indicates whether the current state is a reset state or not) and define $s \models \mathsf{L}_1 \wedge \mathsf{L}_2$ iff $s \models \mathsf{L}_1$ and $s \models \mathsf{L}_2$ for any $s \in \mathcal{S}$, then

- *CartPole*: $\varphi = \neg \mathsf{Reset}\, \mathcal{U}\, \mathsf{Unsafe}$, where $\mathsf{Unsafe} = \langle 1, 1, 0 \rangle$
- *MountainCar*: $\varphi = \neg \mathsf{Goal}\, \mathcal{U}\, \mathsf{Reset}$, where $\mathsf{Goal} = \langle 1, 0, 0, 0 \rangle$
- *Acrobot*: $\varphi = \neg \mathsf{Goal}\, \mathcal{U}\, \mathsf{Reset}$, where $\mathsf{Goal} = \langle 1, 0, \dots, 0 \rangle$
- *LunarLander*: $\varphi = \neg \mathsf{SafeLanding}\, \mathcal{U}\, \mathsf{Reset}$, where $\mathsf{SafeLanding} = \mathsf{GroundContact} \wedge \mathsf{MotorsOff}$, $\mathsf{GroundContact} = \langle 0, 1, 0, 0, 0, 0, 0, 0 \rangle$, and $\mathsf{MotorsOff} = \langle 0, 0, 0, 0, 0, 1, 0 \rangle$
- *Pendulum*: $\varphi = \Diamond(\neg \mathsf{Safe} \wedge \bigcirc \mathsf{Reset})$, where $\mathsf{Safe} = \langle 1, 0, 0, 0, 0 \rangle$, $\Diamond \mathsf{T} = \neg \varnothing\, \mathcal{U}\, \mathsf{T}$, and $s_i \models \bigcirc \mathsf{T}$ iff $s_{i+1} \models \mathsf{T}$, for any $\mathsf{T} \subseteq \mathbf{AP}$, $s_{i:\infty}, a_{i:\infty} \in \textit{Traj}$. Intuitively, $\varphi$ denotes the event of ending an episode in an unsafe state, just before resetting the environment, which means that either the agent never reached the safe region or it reached and left it at some point. Formally, $\varphi = \{ s_{0:\infty}, a_{0:\infty} \mid \exists i \in \mathbb{N}, s_i \not\models \mathsf{Safe} \wedge s_{i+1} \models \mathsf{Reset} \} \subseteq \textit{Traj}$.

## C  ON THE CURSE OF VARIATIONAL MODELING

*Posterior collapse* is a well known issue occurring in variational models (see, e.g., Alemi et al. 2018; Tolstikhin et al. 2018; He et al. 2019; Dong et al. 2020) which intuitively results in a degenerate local optimum where the model learns to ignore the latent space and use only the reconstruction functions (i.e., the decoding distribution) to optimize the objective. VAE-MDPs are no exception, as pointed out in the original paper (Delgrange et al., 2022, Section 4.3 and Appendix C.2).

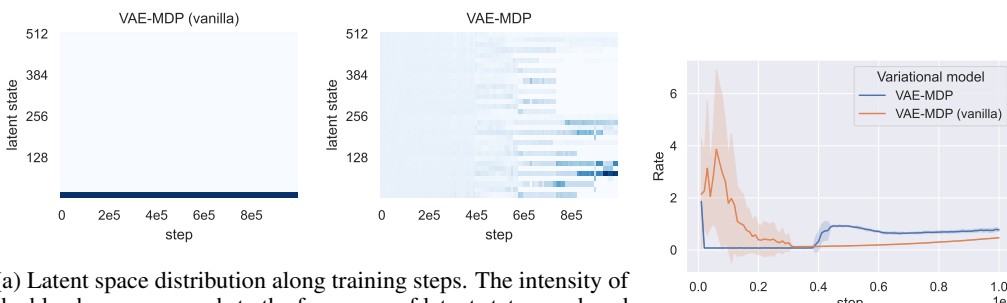

(a) Latent space distribution along training steps. The intensity of the blue hue corresponds to the frequency of latent states produced from $\phi_\iota$ during training. The vanilla model collapses to a single state.

(b) Rate of the variational model.

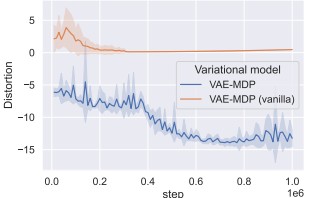 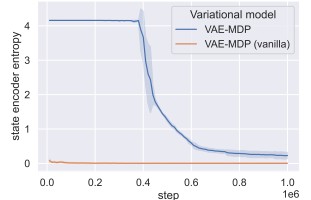 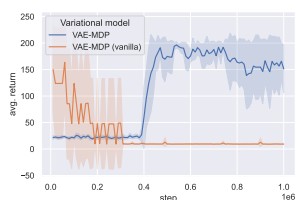

(c) Distortion of the variational model.

(d) Average point-wise entropy of $\phi_\iota(\cdot \mid s)$, for $s \in \mathcal{S}$ drawn from the interaction with the original environment.

(e) Performance of the resulting distilled policy $\bar{\pi}_\theta$ when deployed in the original environment (averaged over 30 episodes).

Figure 7: Comparison of the VAE-MDP in the CartPole environment (i) when the distortion and the rate are minimized as is (*vanilla model*) and (ii) when it makes use of annealing schemes, entropy regularization, and prioritized experience replay to avoid posterior collapse (cf. Delgrange et al. 2022). While the former clearly fails to learn a useful latent representation, the later does so meticulously and smoothly in two distinguishable phases: first, $\phi_\iota$ focuses on fairly distributing the latent space, setting up the stage to the concrete optimization occurring from step $4 \cdot 10^5$, where the entropy of $\phi_\iota$ is lowered, which allows to get the rate of the variational model away from zero. Five instances of the models are trained with different random seeds, with the same hyperparameters than in Sect. 4.

Formally, VAE- and WAE-MDPs optimize their objective by minimizing two losses: a *reconstruction cost* plus a *regularizer term* which penalizes a discrepancy between the encoding distribution and the dynamics of the latent space model. In VAE-MDPs, the former corresponds to the the *distortion*, and the later to the *rate* of the variational model (further details are given in Alemi et al. 2018; Delgrange et al. 2022), while in our WAE-MDPs, the former corresponds to the raw transition distance and the later to both the steady-state and transition regularizers. Notably, the rate minimization of VAE-MDPs involves regularizing a *stochastic* embedding function $\phi_\iota(\cdot \mid s)$ *point-wise*, i.e., for all different input states $s \in \mathcal{S}$ drawn from the interaction with the original environment. In contrast, the latent space regularization of the WAE-MDP involves the marginal embedding distribution $Q_\iota$ where the embedding function $\phi_\iota$ is not required to be stochastic. Alemi et al. (2018) showed that *posterior collapse occurs in VAEs when the rate of the variational model is close to zero,* leading to low-quality representation.

**Posterior collapse in VAE-MDPs.** We illustrate the sensitivity of VAE-MDPs to the posterior collapse problem in Fig. 7, through the CartPole environment[3]: minimizing the distortion and the rate as is yields an embedding function which maps deterministically every input state to the same *sink* latent state (cf. Fig. 7a). Precisely, there is a latent state $\bar{s} \in \overline{\mathcal{S}}$ so that $\phi_\iota(\bar{s} \mid s) \approx 1$ and $\overline{\mathbf{P}}_\theta(\bar{s} \mid \bar{s}, \bar{a}) \approx 1$ whatever the state $s \in \mathcal{S}$ and action $\bar{a} \in \overline{\mathcal{A}}$. This is a form of posterior collapse, the resulting rate quickly drops to zero (cf. Fig 7b), and the resulting latent representation yields no information at all. This phenomenon is handled in VAE-MDPs by using (i) prioritized replay buffers that allow to focus on inputs that led to bad representation, and (ii) modifying the objective

---

[3]In fact, the phenomenon of collapsing to few state occurs for all the environments considered in this paper when their prioritized experience replay is not used, as illustrated in Delgrange et al., 2022, Appendix C.2.

function for learning the latent space model — the so-called evidence lower bound (Hoffman et al., 2013; Kingma & Welling, 2014), or ELBO for short — and set up annealing schemes to eventually recover the ELBO at the end of the training process. Consequently, the resulting learning procedure focuses primarily on fairly distributing the latent space, to avoid it to collapse to a single latent state, to the detriment of learning the dynamics of the environment and the distillation of the RL policy. Then, the annealing scheme allows to make the model learn to finally smoothly use the latent space to maximize the ELBO, and achieve consequently a lower distortion at the "price" of a higher rate.

**Impact of the resulting learning procedure.** The aforementioned annealing process, used to avoid that every state collapses to the same representation, possibly induces a high entropy embedding function (Fig. 7d), which further complicates the learning of the model dynamics and the distillation in the first stage of the training process. In fact, in this particular case, one can observe that the entropy reaches its maximal value, which yields a fully random state embedding function. Recall that the VAE-MDP latent space is learned through *independent* Bernoulli distributions. Fig. 7d reports values centered around $4.188$ in the first training phase, which corresponds to the entropy of the state embedding function when $\phi_\iota(\cdot \mid s)$ is uniformly distributed over $\overline{\mathcal{S}}$ for any state $s \in \mathcal{S}$:
$H(\phi_\iota(\cdot \mid s)) = \sum_{i=0}^{\log_2 |\overline{\mathcal{S}}| - |\mathbf{AP}| = 6} -p_i \log p_i - (1 - p_i) \log(1 - p_i) = 4.188$, where $p_i = \frac{1}{2}$ for all $i$. The rate (Fig. 7b) drops to zero since the divergence pulls the latent dynamics towards this high entropy (yet another form of posterior collapse), which hinders the latent space model to learn a useful representation. However, the annealing scheme increases the rate importance along training steps, which enables the optimization to eventually leave this local optimum (here around $4 \cdot 10^5$ training steps). This allows the learning procedure to leave the zero-rate spot, reduce the distortion (Fig. 7c), and finally distill the original policy (Fig. 7e).

As a result, the whole engineering required to mitigate posterior collapse slows down the training procedure. This phenomenon is reflected in Fig. 4: VAE-MDPs need several steps to stabilize and set up the stage to the concrete optimization, whereas WAE-MDPs have no such requirements since they naturally do not suffer from collapsing issues (cf. Fig. 5), and are consequently faster to train.

**Lack of representation guarantees.** On the theoretical side, since VAE-MDPs are optimized via the ELBO and the local losses via the related variational proxies, VAE-MDPs *do not leverage the representation quality guarantees* induced by local losses (Eq. 1) during the learning procedure (as explicitly pointed out by Delgrange et al., 2022, Sect. 4.1.): in contrast to WAE-MDPs, when two original states are embedded to the same latent, abstract state, the former are not guaranteed to be bisimilarly close (i.e., the agent is not guaranteed to behave the same way from those two states by executing the policy), meaning those proxies do not prevent original states having distant values collapsing together to the same latent representation.

## INDEX OF NOTATIONS

$\mathbf{1}_{[cond]}$     indicator function: $1$ if the statement $[cond]$ is true, and $0$ otherwise

$\mathcal{F}_d$       Set of 1-Lipschitz functions w.r.t. the distance metric $d$

$\sigma$         Sigmoid function, with $\sigma(x) = \frac{1}{1 + \exp(-x)}$

$f_\theta$        A function $f_\theta \colon \mathcal{X} \to \mathbb{R}$ modeled by a neural network, parameterized by $\theta$, where $\mathcal{X}$ is any measurable set

**Latent Space Model**

$\overline{\mathcal{M}} = \langle \overline{\mathcal{S}}, \overline{\mathcal{A}}, \overline{\mathbf{P}}, \overline{\mathcal{R}}, \bar{\ell}, \mathbf{AP}, \bar{s}_I \rangle$   Latent MDP with state space $\overline{\mathcal{S}}$, action space $\overline{\mathcal{A}}$, reward function $\overline{\mathcal{R}}$, labeling function $\bar{\ell}$, atomic proposition space $\mathbf{AP}$, and initial state $\bar{s}_I$.

$\langle \overline{\mathcal{M}}, \phi, \psi \rangle$   Latent space model of $\mathcal{M}$

$\bar{a}$        Latent action in $\overline{\mathcal{A}}$

$\bar{\pi}$        Latent policy $\bar{\pi} \colon \overline{\mathcal{S}} \to \mathcal{A}$; can be executed in $\mathcal{M}$ via $\phi$: $\bar{\pi}(\cdot \mid \phi(s))$

$d_{\overline{\mathcal{S}}}$       Distance metric over $\overline{\mathcal{S}}$

$\phi$        State embedding function, from $\mathcal{S}$ to $\overline{\mathcal{S}}$

$\psi$      Action embedding function, from $\bar{\mathcal{S}} \times \overline{\mathcal{A}}$ to $\mathcal{A}$

$\phi\mathbf{P}$      Distribution of drawing $s' \sim \mathbf{P}(\cdot \mid s, a)$, then embedding $\bar{s}' = \phi(s')$, for any state $s \in \mathcal{S}$ and action $a \in \mathcal{A}$

$L_{\mathcal{R}}^{\xi}$      Local reward loss under distribution $\xi$

$L_{\mathbf{P}}^{\xi}$      Local transition loss under distribution $\xi$

$\overline{\Pi}$      Set of (memoryless) latent policies

$\bar{s}$      Latent state in $\bar{\mathcal{S}}$

$\overline{V}_{\bar{\pi}}^{\cdot}$      Latent value function

**Markov Decision Processes**

$\mathcal{M} = \langle \mathcal{S}, \mathcal{A}, \mathbf{P}, \mathcal{R}, \ell, \mathbf{AP}, s_I \rangle$ MDP $\mathcal{M}$ with state space $\mathcal{S}$, action space $\mathcal{A}$, transition function $\mathbf{P}$, labeling function $\ell$, atomic proposition space $\mathbf{AP}$, and initial state $s_I$.

$a$      Action in $\mathcal{A}$

$\widetilde{d}_{\pi}$      Bisimulation pseudometric

$\gamma$      Discount factor in $[0, 1]$

$d_{\mathcal{A}}$      Metric over the action space

$d_{\mathcal{R}}$      Metric over $\mathrm{Im}(\mathcal{R})$

$d_{\mathcal{S}}$      Metric over the state space

$\xi_{\pi}^{t}$      Limiting distribution of the MDP defined as $\xi_{\pi}^{t}(s' \mid s) = \mathbb{P}_{\pi}^{\mathcal{M}_s}\left(\{ s_{0:\infty}, a_{0:\infty} \mid s_t = s' \}\right)$, for any source state $s \in \mathcal{S}$

$\Pi$      Set of memoryless policies of $\mathcal{M}$

$\pi$      Memoryless policy $\pi \colon \mathcal{S} \to \Delta(\mathcal{A})$

$\mathbb{P}_{\pi}^{\mathcal{M}}$      Unique probability measure induced by the policy $\pi$ in $\mathcal{M}$ on the Borel $\sigma$-algebra over measurable subsets of $Traj$

$\mathsf{C}\,\mathcal{U}\,\mathsf{T}$      Constrained reachability event

$\mathcal{M}_s$      MDP obtained by replacing the initial state of $\mathcal{M}$ by $s \in \mathcal{S}$

$s$      State in $\mathcal{S}$

$\xi_{\pi}$      Stationary distribution of $\mathcal{M}$ induced by the policy $\pi$

$\vec{d}$      Raw transition distance, i.e., metric over $\mathcal{S} \times \mathcal{A} \times \mathrm{Im}(\mathcal{R}) \times \mathcal{S}$

$Traj$      Set of infinite trajectories of $\mathcal{M}$

$\tau = \langle s_{0:T}, a_{0:T-1} \rangle$ Trajectory

$V_{\pi}^{\cdot}$      Value function for the policy $\pi$

**Probability / Measure Theory**

$D$      Discrepancy measure; $D(P, Q)$ is the discrepancy between distributions $P, Q \in \Delta(\mathcal{X})$

$\Delta(\mathcal{X})$      Set of measures over a complete, separable metric space $\mathcal{X}$

$\mathrm{Logistic}(\mu, s)$      Logistic distribution with location parameter $\mu$ and scale parameter $s$

$W_d$      Wasserstein distance w.r.t. the metric $d$; $W_d(P, Q)$ is the Wasserstein distance between distributions $P, Q \in \Delta(\mathcal{X})$

**Wasserstein Auto-encoded MDP**

$\xi_{\theta}$      Behavioral model: distribution over $\mathcal{S} \times \mathcal{A} \times \mathrm{Im}(\mathcal{R}) \times \mathcal{S}$

$G_{\theta}$      Mapping $\langle \bar{s}, \bar{a}, \bar{s}' \rangle \mapsto \langle \mathcal{G}_{\theta}(\bar{s}), \psi_{\theta}(\bar{s}, \bar{a}), \overline{\mathcal{R}}_{\theta}(\bar{s}, \bar{a}), \mathcal{G}_{\theta}(\bar{s}') \rangle$

$\phi_{\iota}^{\mathcal{A}}$      Action encoder mapping $\bar{\mathcal{S}} \times \mathcal{A}$ to $\Delta(\overline{\mathcal{A}})$

$\mathcal{G}_{\theta}$      State-wise decoder, from $\bar{\mathcal{S}}$ to $\mathcal{S}$

$Q_\iota$      Marginal encoding distribution over $\bar{\mathcal{S}} \times \bar{\mathcal{A}} \times \bar{\mathcal{S}} : \mathbb{E}_{s,a,s' \sim \xi_\pi} \phi_\iota(\cdot \mid s, a, s')$

$\bar{\xi}_{\bar{\pi}_\theta}$      Stationary distribution of the latent model $\overline{\mathcal{M}}_\theta$, parameterized by $\theta$

$\mathcal{W}_{\xi_{\bar{\pi}}}$      Steady-state regularizer

$\varphi_\omega^\xi$      Steady-state Lipschitz network

$\lambda$      Temperature parameter

$\mathcal{T}$      Distribution of drawing state-action pairs from interacting with $\mathcal{M}$, embedding them to the latent spaces, and finally letting them transition to their successor state in $\overline{\mathcal{M}}_\theta$, in $\Delta(\bar{\mathcal{S}} \times \bar{\mathcal{A}} \times \bar{\mathcal{S}})$

$\varphi_\omega^{\mathbf{P}}$      Transition Lipschitz network

## ADDITIONAL REFERENCES

Martín Abadi, Ashish Agarwal, Paul Barham, Eugene Brevdo, Zhifeng Chen, Craig Citro, Greg S. Corrado, Andy Davis, Jeffrey Dean, Matthieu Devin, Sanjay Ghemawat, Ian Goodfellow, Andrew Harp, Geoffrey Irving, Michael Isard, Yangqing Jia, Rafal Jozefowicz, Lukasz Kaiser, Manjunath Kudlur, Josh Levenberg, Dandelion Mané, Rajat Monga, Sherry Moore, Derek Murray, Chris Olah, Mike Schuster, Jonathon Shlens, Benoit Steiner, Ilya Sutskever, Kunal Talwar, Paul Tucker, Vincent Vanhoucke, Vijay Vasudevan, Fernanda Viégas, Oriol Vinyals, Pete Warden, Martin Wattenberg, Martin Wicke, Yuan Yu, and Xiaoqiang Zheng. TensorFlow: Large-scale machine learning on heterogeneous systems, 2015. URL https://www.tensorflow.org/. Software available from tensorflow.org.

Alexander A. Alemi, Ben Poole, Ian Fischer, Joshua V. Dillon, Rif A. Saurous, and Kevin Murphy. Fixing a broken ELBO. In Jennifer G. Dy and Andreas Krause (eds.), *Proceedings of the 35th International Conference on Machine Learning, ICML 2018, Stockholmsmässan, Stockholm, Sweden, July 10-15, 2018*, volume 80 of *Proceedings of Machine Learning Research*, pp. 159–168. PMLR, 2018. URL http://proceedings.mlr.press/v80/alemi18a.html.

Joshua V. Dillon, Ian Langmore, Dustin Tran, Eugene Brevdo, Srinivas Vasudevan, Dave Moore, Brian Patton, Alex Alemi, Matt Hoffman, and Rif A. Saurous. Tensorflow distributions, 2017.

Zhe Dong, Bryan A. Seybold, Kevin Murphy, and Hung H. Bui. Collapsed amortized variational inference for switching nonlinear dynamical systems. In *Proceedings of the 37th International Conference on Machine Learning, ICML 2020, 13-18 July 2020, Virtual Event*, volume 119 of *Proceedings of Machine Learning Research*, pp. 2638–2647. PMLR, 2020. URL http://proceedings.mlr.press/v119/dong20e.html.

Sergio Guadarrama, Anoop Korattikara, Oscar Ramirez, Pablo Castro, Ethan Holly, Sam Fishman, Ke Wang, Ekaterina Gonina, Neal Wu, Efi Kokiopoulou, Luciano Sbaiz, Jamie Smith, Gábor Bartók, Jesse Berent, Chris Harris, Vincent Vanhoucke, and Eugene Brevdo. TF-Agents: A library for reinforcement learning in tensorflow. https://github.com/tensorflow/agents, 2018. URL https://github.com/tensorflow/agents. [Online; accessed 25-June-2019].

Junxian He, Daniel Spokoyny, Graham Neubig, and Taylor Berg-Kirkpatrick. Lagging inference networks and posterior collapse in variational autoencoders. In *7th International Conference on Learning Representations, ICLR 2019, New Orleans, LA, USA, May 6-9, 2019*. OpenReview.net, 2019. URL https://openreview.net/forum?id=rylDfnCqF7.

Matthew D. Hoffman, David M. Blei, Chong Wang, and John W. Paisley. Stochastic variational inference. *J. Mach. Learn. Res.*, 14(1):1303–1347, 2013. URL http://dl.acm.org/citation.cfm?id=2502622.

Diederik P. Kingma and Jimmy Ba. Adam: A method for stochastic optimization. In Yoshua Bengio and Yann LeCun (eds.), *3rd International Conference on Learning Representations, ICLR 2015, San Diego, CA, USA, May 7-9, 2015, Conference Track Proceedings*, 2015. URL http://arxiv.org/abs/1412.6980.

Diederik P. Kingma and Max Welling. Auto-encoding variational bayes. In Yoshua Bengio and Yann LeCun (eds.), *2nd International Conference on Learning Representations, ICLR 2014, Banff, AB, Canada, April 14-16, 2014, Conference Track Proceedings*, 2014. URL `http://arxiv.org/abs/1312.6114`.

Vidyadhar G. Kulkarni. *Modeling and Analysis of Stochastic Systems*. Chapman & Hall, Ltd., GBR, 1995. ISBN 0412049910.

Ilya O. Tolstikhin, Olivier Bousquet, Sylvain Gelly, and Bernhard Schölkopf. Wasserstein auto-encoders. In *6th International Conference on Learning Representations, ICLR 2018, Vancouver, BC, Canada, April 30 - May 3, 2018, Conference Track Proceedings*. OpenReview.net, 2018. URL `https://openreview.net/forum?id=HkL7n1-0b`.

