# OpenReview forum: "Wasserstein Auto-encoded MDPs: Formal Verification of Efficiently Distilled RL Policies with Many-sided Guarantees"
_ICLR.cc/2023/Conference — ICLR 2023 poster_

### Official Review · Reviewer_MegB · 2022-10-21

**Confidence:** 2
**Correctness:** 3
**Technical Novelty And Significance:** 2
**Empirical Novelty And Significance:** 2
**Recommendation:** 5

**Clarity, Quality, Novelty And Reproducibility:**

In my view, clarity is where the paper immediately falls short. It is extremely difficult to follow if you do not work in that specific area. Quality- and novelty-wise I believe the paper may perform well. In terms of representation, the authors provide the source code of their implementation and notebook videos of how the policies offered by the proposed approach behave in practice.

**Strength And Weaknesses:**

I should admit that I am no expert in this particular area and so I may have overlooked some of the important details of the work.

Having said that, I should bring the authors' attention to the fact that the paper is not helping much in this regard. Although it has a section on "background", in my opinion, it still does extremely little when introducing preliminary knowledge - the authors cut to the chase and start throwing bulky "two-storey" formulas, equations, and theorems with no motivation whatsoever. It may be okay for an expert in the field, e.g. the authors themselves, but an average reader like me will get lost immediately. The algorithm description provided follows the same style, which makes it really hard to appreciate the presentation provided.

In my view, "formal verification" means logical reasoning about a system and formally proving certain properties of the system. This paper does not make it clear what kind of reasoning engine is applied when verifying the properties. I should also say that it does not discuss the practicality of the proposed approach (say, from the scalability perspective).

**Summary Of The Paper:**

This paper is devoted to verifying deep reinforcement learning (DRL) policies and proposes a latent space model (Wasserstein auto-encoded MDP, WAE-MDP) that addresses a number of issues pertaining to earlier latent space models by minimising the optimal transport between the behaviours of the agent executing the original policy and the distilled policy. Here, formal guarantees for the distilled policy are provided. Experimental results demonstrate the proposed approach to be faster in distilling policies and better in terms of the overall result quality.

**Summary Of The Review:**

As I feel completely lost in the text, I am afraid I cannot give this paper a positive score. However, as I mentioned above, I could have overlooked important merits of the paper, hence, my score is "borderline reject".

---

> ### Author Response · Authors · 2022-11-10
> **Response to Reviewer MegB**
>
> We regret that the reviewer finds the article difficult to read, but we understand that he or she is not an expert in this area. We hope we can clarify some of their concerns.
>
> **On formal verification:** Formal verification techniques range from fully automatic to manual (mostly theorem provers). We believe the reviewer refers to the latter as they mention that verification requires a reasoning engine to establish properties about a system. In this work, we are mostly interested in the first kind of verification techniques: fully-automated verification of MDPs and policies (as models of a system) against a formal specification (which can take the form of, e.g. a linear temporal logic (LTL) formula, just as objectives are sometimes specified in the planning community). One such technique is model checking, which was popularized by Clarke, Emerson, and Sifakis who received a Turing award for it. In layman’s terms: a model checking tool takes the MDP, a policy, and an LTL formula and will output yes or no depending on whether the resulting Markov chain satisfies the formula. The main restriction for the application of model checking on a learned policy from a (partially known, and huge) continuous MDP is indeed scalability as rightfully pointed out by the reviewer. More importantly, the fact that the MDP might be unknown also hampers the application of model checking which requires a fully known model. Now, our work goes beyond merely providing a simplified version of the MDP underlying the environment on which we can learn, as well as a simplified version of the learned policy. That is, we even provide formal guarantees concerning the relation between the learned objects and the ones we learn from. These (bisimulation) guarantees make it so that if we model check the learned objects, then the yes/no answer from a model-checking tool, is (approximately) applicable to the unknown environment! As such, our main contribution is in providing a framework which allows to learn these carefully simplified MDP and carefully simplified policy while proving that the bisimulation guarantees do indeed hold (a rather theoretical task thus).
>
> Hopefully, the context is now clearer to the reviewer. We apologize for the amount of notation we required to present the formal guarantees (one of our main contributions). While we do not want to compromise the correctness of these guarantees and the statements we make about them, other reviewers have provided us with nice recommendations to improve their presentation to not scare average readers. We will also include all of the above information (regarding the context of our work) in our final version of the paper and will stay mindful of interested readers that may not have knowledge of verification or formal reasoning of MDPs.

---

### Official Review · Reviewer_67Kn · 2022-10-24

**Confidence:** 3
**Correctness:** 3
**Technical Novelty And Significance:** 3
**Empirical Novelty And Significance:** 3
**Recommendation:** 6

**Clarity, Quality, Novelty And Reproducibility:**

The paper is quite dense, with a lot of dependency on the appendix.  Many of the proofs of the critical claims are in the appendix. Graphs in Figure 4 are very difficult to read. Some clarification on novelty of the paper is requested in the Weaknesses section of the paper.

**Strength And Weaknesses:**

Strengths:

+ The paper presents a framework for learning RL policy distillations with bisimulation guarantees. This enables formal verification over learned discrete abstraction of continuous environment. This approach improves upon VAE-MDP in terms of learning speed and model performance.

Weaknesses:

- While the paper talks about formal verification, it fails to clearly identify the kind of formal guarantees it aims to establish beyond bisimulation and reachability guarantees. While it makes a theoretical claim of verifying general discounted omega-regular properties, the experimental evaluation fall far short of it.

- The experimental results are limited to toy examples - cartpole, pendulum, mountain car, acrobot and lunarlander. It would be helpful to consider more complex RL environment such as highD control examples.

**Summary Of The Paper:**

The paper learns a discrete abstraction of the state-action space of the MDP underlying an RL setup. This abstraction has smaller latent space and the distilled RL policy is more tractable for formal verification approaches such as model checking of bisimulation guarantees. The paper uses Waserstien autoencoders to overcome several limitations with the use of VAEs and builds on recent work by Delgrange et al in AAAI'22.

**Summary Of The Review:**

This paper might be more suitable for COLT, SODA or STOCS. It is not clear if this paper has enough novelty and relevance to a machine learning audience. The paper is mostly about MDP abstraction. In addition to clarifying the weaknesses identified in this review, the authors are requested to draw a stronger connection to RL. The mismatch in theory and experiments in the paper suggest that this is a work in progress.

---

> ### Author Response · Authors · 2022-11-10
> **Response to Reviewer 67Kn**
>
> **Novelty.** We respectfully disagree with the reviewer’s statement that the paper might not have enough relevant contributions to the ICLR audience. We note that the list of relevant topics of the conference includes “theoretical issues in deep learning”. Our contribution is indeed in the direction of new RL algorithms with correctness guarantees.
>
> *We will make the following points clearer in the final version of our manuscript.* \
> **Formal verification.** *Our bisimulation guarantees come by construction of the latent space.* Essentially, our learning algorithm spits out a distilled policy and a latent state space which already yields a guaranteed bisimulation distance between the original MDP and the latent MDP. This is the crux of how we enable verification techniques like model checking, even if we do not study them or benchmark them directly. In particular, bisimulation guarantees mean that reachability probabilities in the latent MDP vs the original one are close. Furthermore, the value difference of (omega-regular) properties (formulated through mu-calculus) obtained in the two models is bounded by this distance (cf. [8]). Reachability is the key ingredient to model-check MDPs. Model-checking properties is in most cases performed by reduction to the reachability of components or regions of the MDP: it either consists of
> (i) iteratively checking the reachability of the parts of the state space satisfying path formulae that comprise the specification, through a tree-like decomposition of the latter (e.g., for (P,R-)CTL properties [9]), or
> (ii)  checking the reachability of the part of the state space of a product of the MDP with a memory structure or an automaton that embeds the omega-regular property (e.g., for LTL [10,11,12], LTLf [13], or GLTL [14], among other specification formalisms).
> The choice of specification formalism is up to the user and depends on the case study. The scope of this work is *focusing on learning to distill reinforcement policies with bisimulation guarantees so that model checking can be applied, in order to reason about the behaviors of the agent.* That being said, reachability is all we need to show that model checking can be applied.
>
> **Experimental evaluation.** Although the environments considered in our work might seem like toy examples for other deep-RL algorithms, to the best of our knowledge, there is no other learning procedure that constructs a simpler version of an unknown environment with continuous spaces, as well as a policy that is amenable to model checking, such that formal guarantees can be provided. In that regard, these are the very first benchmarks for (deep) RL for which such formal guarantees are provided. We thus believe that those environments are relevant and challenging to our approach. As a concrete example, LunarLander is a difficult environment to approach through fully discretized spaces on which, very often, RL algorithms fail to control the system. Also the Pendulum environment has non-trivial properties to check, for example, whether the agent is able to raise the pendulum and then keep it in balance (which we have formally checked, details are given in Appendix B.5). We stress that our focus is not on the challenge of learning the RL policy; Deep-RL has already shown to be capable of solving high-dimensional control scenarios, this is not our contribution. Our research aims at providing guarantees, which are currently lacking.
> Considering a challenging real-world scenario, where verifying critical, concrete, and interesting properties is part of our plans for future work.
>
> **Dependency on the Appendix.** We indeed moved to the Appendix the notions and full proofs that are not essential for understanding the paper, but importantly, we also included in the main text the intuitions behind each claim, as well as Figures to complement the text (as kindly pointed out by reviewer JqQE). We stress that a large part of our contribution consists in new theories about formal guarantees. We understand that some parts of the paper are denser than others, but we have to balance this with not losing the correctness of our claims. The other reviewers proposed nice ideas to improve on this point, that we will implement for the final version.
>
> We hope that our answer clarified the weaknesses pointed out and did convince the reviewer on our motivation and evaluations, as we did for reviewers cw5C and JqQE (who is confident on his or her assessment).

---

> > ### Author Response · Authors · 2022-11-10
> > **References**
> >
> > [8] Krishnendu Chatterjee, Luca de Alfaro, Rupak Majumdar, Vishwanath Raman: Algorithms for Game Metrics (Full Version). Log. Methods Comput. Sci. 6(3) (2010) \
> > [9] Christel Baier, Joost-Pieter Katoen: Principles of model checking. MIT Press 2008, ISBN 978-0-262-02649-9, pp. I-XVII, 1-975 \
> > [10] Christel Baier, Stefan Kiefer, Joachim Klein, Sascha Klüppelholz, David Müller, James Worrell: Markov Chains and Unambiguous Büchi Automata. CAV (1) 2016: 23-42 \
> > [11] Salomon Sickert, Javier Esparza, Stefan Jaax, Jan Kretínský: Limit-Deterministic Büchi Automata for Linear Temporal Logic. CAV (2) 2016: 312-332 \
> > [12] Alper Kamil Bozkurt, Yu Wang, Michael M. Zavlanos, Miroslav Pajic: Control Synthesis from Linear Temporal Logic Specifications using Model-Free Reinforcement Learning. ICRA 2020: 10349-10355 \
> > [13] Andrew M. Wells, Morteza Lahijanian, Lydia E. Kavraki, Moshe Y. Vardi: LTLf Synthesis on Probabilistic Systems. GandALF 2020: 166-181 \
> > [14] Michael L. Littman, Ufuk Topcu, Jie Fu, Charles Lee Isbell Jr., Min Wen, James MacGlashan: Environment-Independent Task Specifications via GLTL. CoRR abs/1704.04341 (2017)

---

> > > ### Comment · Reviewer_67Kn · 2022-11-30
> > > **Thank you**
> > >
> > > The main concerns of the reviewer have been met, and the reviewer has raised the rating.

---

### Official Review · Reviewer_JqQE · 2022-10-24

**Confidence:** 4
**Correctness:** 4
**Technical Novelty And Significance:** 3
**Empirical Novelty And Significance:** 3
**Recommendation:** 8

**Clarity, Quality, Novelty And Reproducibility:**

The paper is quite clear and of high quality. The method presented is novel, as far as I can tell.
The authors have included technical details for their method, and have provided source code for reproducibility.

**Strength And Weaknesses:**

# Strengths
The paper is well written, well motivated, and properly evaluated. The theoretical claims are presented clearly and the proofs are correct (as far as I could tell).

I wanted to commend the authors on Figures 1 and 2, which really nicely complement the text and help clarify the methods discussed.

# Weaknesses
The main weakness for me is the complexity of the method presented, although that is difficult to resolve. The paper is quite clear, but there are a few points where it could be made clearer:
1. There is a _lot_ of notation. It would be useful to provide an index of notation inthe appendix, as it can be hard to keep track of all of them.
1. In the top of page 4, it is not clear whether $\bar{\mathcal{R}}$ and $\bar{\bf{P}}$ are learned or not. I think they are, but this should be made explicit. In particular, they should include the parameterization $\theta$ (which is present in Figure 2) to make it clearer.
1. In section 3.1, shouldn't there be an assumption on the scales of the metrics $d_S$, $d_A$, and $d_R$ being similar? Otherwise simply adding them doesn't make sense.
1. Below Theorem 3.1 there is an objective presented. If I understand correctly that objective is the loss version of the equation in Thm 3.1. If so, what's the motivation for the last ($\beta$-weighted) term?

# Questions / suggestions
1. In section 2 MDPs are introduced as having a single initial state $s_I$. Do the results not hold if one has an initial state distribution instead?
1. In the bottom of page 2 it says $s\models \top$ if $\ell(s)\cap \top \ne\emptyset$. Should it not be $\top\subseteq\ell(s)$ instead? In the form presented $s\models \top$ as long as at least _one_ **AP** in $\top$ is in $\ell(s)$, which I don't think is what is meant.
1. In the **Bisimulation** section of page 4, what is meant by "the largest bisimulation relation"?
1. In the paragraph above Lemma 3.2 it says "along with setting $D$ to $W_{\overrightarrow{d}}$, yield ...". This suggests your loss is somehow computing $(1 + \beta)W_{\overrightarrow{d}}$, or am I missing something?
1. In Figure 4(c), why are no learning curves shown for RL policy? Where are the lines for original (DQN) and original (SAC)?
1. In the paragraph below Algorithm 1 it appears the indicators "former" and "latter" are swapped.
1. The authors use the term "Gumble", but it should be "Gumble".
1. In the **Latent distributions** paragraph on page 7, it should read "We emphasi**ze** that this trick alone..."
1. In the first paragraph of the conclusion it should say "Our method overcome**s** the limitations..."

**Summary Of The Paper:**

This paper introduces the Wasserstein auto-encoded MDP (WAE-MDP), which is a latent space model that aims to overcome some of the shortcomings of a prior latent space model (VAE-MDP). Their method learns a (small) discrete representation of the state-action space. Their approach lends itself to formal guarantees on behaviour (based on the theory of bisimulation). In addition to enabling formal guarantees, the authors demonstrate their method is 10 times faster than VAE-MDP in distilling policies.

**Summary Of The Review:**

A nice paper that provides a theoretically-motivated methodology for learning a latent space model on which formal verification can be performed.

I support accepting the paper, provided the clarity points above are addressed.

---

> ### Author Response · Authors · 2022-11-10
> **Response to Reviewer JqQE**
>
> ### Response on weaknesses
>
> 1. Thank you for the recommendation, this is indeed a really nice idea. We will add such a notation table to the Appendix.
> 2. Indeed, our final goal is to learn all the model components (as mentioned in the “latent MDP” paragraph of page 3) through parameterized neural networks. The guarantees presented in the background section hold for any general latent space model, regardless of the method used to obtain it. Therefore, we did not include them to not overload the notations. However, we’ll make sure to mention clearly that our goal is to learn those functions in the final version of the paper.
> 3. Note that $\overrightarrow{d}$, which consists of the sum of the distance of all the transition components, is a sound distance, whatever $d_S$, $d_A$, $d_R$, since the sum of distances preserves the identity of indiscernible, symmetry, and triangle inequality. However, in practice, the choice of the distance function plays indeed a role in the success of our approach. The usual L2 distance is often a good choice for all the transition components, but the scale, dimensionality, and nature of the inputs sometimes require using scaled, normalized, or other kinds of distances to allow the network to reconstruct each component. While we did not observe such requirements in our experiments (where we simply used the L2 distance), high dimensional observations (e.g., images) are an example of data which could require tuning the state-distance function in such a way, to make sure that the optimization of the reward or action reconstruction will not be disfavored compared to that of the states.
> 4. (the following also answers point 4 in the Questions) Indeed, the objective is the loss version of the equation presented in Theorem 3.1. In this equation, notice that the infimum is taken over the marginal encoding distributions ($Q$) that match the prior (the latent stationary distribution $\bar{\xi}\_\bar{\pi}$ ). In the loss version, we enforce $Q$ to match $\bar{\xi}_{\bar{\pi}}$ by minimizing a discrepancy $D$ between the two. When $D$ is chosen to be the Wasserstein distance again (but this time for regularizing the latent space rather than learning a behavioral model of the original environment), we derive an upper bound on the objective which directly incorporates the local transition loss, as well as a steady-state regularizer that enables the distillation (Lemma 3.2).
>
> ### Response to the Questions
>
> 1. Yes, indeed, the results remain unchanged. Intuitively, assume the model has an initial distribution $\delta_I$, then it suffices to add a dummy state $s_{I}$ to the state space with the following transition dynamics: $\mathbf{P}(s' \mid s_I, a) = \delta_I(s')$ for any state $s'$ and action $a$. Therefore, each time the reset of the environment is triggered, we make the MDP entering $s_I$, then transitioning to $s'$ according to $\delta_I$.
> 2. No, this is actually exactly what we mean: assume one desires to verify that the agent is reaching a target state, and that we capture this target through the atomic proposition “goal”. Then we consider that every state labeled with “goal” is indeed a target state, regardless of the other atomic propositions in its label. However, one can add a dependency on the other atomic propositions by adding a “$\wedge$” operator, as we did in Appendix B.5 (at the bottom of page 22).
> 3. We are interested in the largest bisimulation relation, or in other words, the bisimulation that leads to the coarsest partition of the state space and groups together the more states: bisimulation properties guarantee that the values of (and traces issued from) states in bisimulation (i.e., in the same equivalence class) are the same, even with the coarsest partition. In contrast, the smallest bisimulation relation, or the bisimulation relation which leads to the finest partition, is the identity with $s_1 \mathcal{B} s_2$ iff $s_1 = s_2$.
> 4. See point 4 above.
> 5. We did not report the learning process of DQN and SAC since we fixed the RL policies to distill them via V-, and WAE-MDPs. The plots of Figure 4c report the distillation of such policies, i.e., the number of steps required to recover their performance through the distillation. The straight green line in the plots reports the performance of the (fixed) original RL policy (so, either a policy issued from DQN or SAC, according to the environment) and are provided for information purposes as a baseline to be met or overcome by the distilled policies.
> 6. 7. 8. 9. Thank you for spotting these typos. We will correct them for the final version of the paper.

---

> > ### Comment · Reviewer_JqQE · 2022-11-14
> > **Thanks**
> >
> > Thank you for the responses to my questions, they've addressed my concerns.

---

### Official Review · Reviewer_cw5C · 2022-10-26

**Confidence:** 2
**Correctness:** 3
**Technical Novelty And Significance:** 3
**Empirical Novelty And Significance:** 3
**Recommendation:** 8

**Clarity, Quality, Novelty And Reproducibility:**

The authors improve VAE-MDP by replacing a VAE with a WAE. The authors proposed a new objective function and provided theories for it. I consider the method novel.
The presentations of the paper are generally straightforward. In Figure 4, does the horizontal axis represents the number of training step?

**Strength And Weaknesses:**

Strengths

The authors proposed a novel method called WAE-MDP.
The authors provided theoretical guarantees for WAE-MDP.
Experimental results show that WAE-MDP outperforms VAE-MDP

Weaknesses

The authors claim that one of the reasons why VAE-MDP does not work well is due to the mode collapse of VAE. However, limited evidence is provided that VAE suffers from mode collapse.

**Summary Of The Paper:**

This paper focuses on improving the performance of deep reinforcement learning. The authors proposed Wasserstein auto-encoded Markov Decision Processes (WAE-MDPs) for solving this problem. Experimental results show that WAE-MDPs outperform VAE-MDPs.

**Summary Of The Review:**

The authors propose a novel method called WAE-MDP. Experimental results show that WAE-MDP outperforms VAE-MDP

---

> ### Author Response · Authors · 2022-11-10
> **Response to Reviewer cw5C**
>
> **Figure 4.** Indeed, the horizontal axis reports the training step of the V- and WAE-MDPs.
>
> **Issues with variational modeling.** Mode collapse is a well known issue occurring in variational models (see, e.g., [1,2,3,4]). VAE-MDPs are no exception, as pointed out in the original paper [5: Section 4.3 and Appendix C.2]. In fact, the authors handle the problem by modifying the objective function for learning the latent space model (the so-called evidence lower bound [6,7], or ELBO for short) and set up annealing schemes to eventually recover the ELBO at the end of the training process. Consequently, the resulting learning procedure focuses primarily on reconstructing the inputs (referred to as the “distortion” minimization) to the detriment of learning the dynamics of the environment and the distillation of the RL policy (via the “rate” minimization). Furthermore, the annealing process also embeds an entropy regularization term on the (stochastic) state embedding function, which aims at fairly distributing the latent space and allows it to avoid mapping every state to the same representation. However, the possibly high entropy of the resulting encoder makes learning the dynamics of the model and the distillation even more difficult in the first stage of the training process. As a result, these slow down the training procedure and this phenomenon is reflected in Figure 4 of our paper: VAE-MDPs need several steps to stabilize and set up the stage to the concrete optimization, whereas WAE-MDPs have no such requirements, and are consequently faster to train.
>
> On the theoretical side, since VAE-MDPs are optimized via the ELBO and the local losses via the related variational proxies, VAE-MDPs do not leverage the representation quality guarantees induced by local losses (Equation 1 in our paper) during the learning procedure: in contrast to WAE-MDPs, when two original states are embedded to the same latent, abstract state, the former are not guaranteed to be bisimilarly close*, meaning those proxies do not prevent original states having distant values collapsing together to the same representation.
>
> *the agent is not guaranteed to behave the same way from those two states by executing the policy.
>
> [1] Alexander A. Alemi, Ben Poole, Ian Fischer, Joshua V. Dillon, Rif A. Saurous, Kevin Murphy: Fixing a Broken ELBO. ICML 2018: 159-168 \
> [2] Junxian He, Daniel Spokoyny, Graham Neubig, Taylor Berg-Kirkpatrick: Lagging Inference Networks and Posterior Collapse in Variational Autoencoders. ICLR 2019 \
> [3] Zhe Dong, Bryan A. Seybold, Kevin Murphy, Hung H. Bui: Collapsed Amortized Variational Inference for Switching Nonlinear Dynamical Systems. ICML 2020: 2638-2647 \
> [4] Ilya O. Tolstikhin, Olivier Bousquet, Sylvain Gelly, Bernhard Schölkopf: Wasserstein Auto-Encoders. ICLR 2018 \
> [5] Florent Delgrange, Ann Nowé, Guillermo A. Pérez: Distillation of RL Policies with Formal Guarantees via Variational Abstraction of Markov Decision Processes. AAAI 2022: 6497-6505. \
> [6] Matthew D. Hoffman, David M. Blei, Chong Wang, John W. Paisley: Stochastic variational inference. J. Mach. Learn. Res. 14(1): 1303-1347 (2013) \
> [7] Diederik P. Kingma, Max Welling: Auto-Encoding Variational Bayes. ICLR 2014

---

> > ### Comment · Reviewer_cw5C · 2022-11-16
> > **Thanks for the response.**
> >
> > Thanks for the response. In the paper, it is claimed that the mode collapse is the reason for the failure of VAE-MDPs. However, there is no experimental evidence to support this claim. Despite that I suggest the acceptance of this paper; I consider this as a limitation of this paper.

---

> > > ### Author Response · Authors · 2022-11-18
> > > **Additional experiments demonstrating collapsing issues in VAE-MDPs**
> > >
> > > Thank you for insisting on this. We have now submitted a revision of our paper including a new section (Appendix C) detailing the collapsing issues, with experiments demonstrating that those occur in the latent space of VAE-MDPs — which is not the case with our approach. This also allowed us to notice that our use of the term *mode* collapse was a misnomer. We actually meant *posterior* collapse. *Mode* collapse refers to a similar problem occurring in GANs, where the generator always produces the same output, or a small subset of outputs, whereas *posterior* collapse refers to the problem where the decoder (the counterpart of the generator) learns to ignore the latent space to optimize the objective function on its own. We corrected this in our revision.

---

### Author Response · Authors · 2022-11-10
**General response**

We thank the reviewers for their insightful comments. We hope to have addressed the reviewers’ concerns and we will follow their recommendations to improve on the clarity when preparing the final version of the paper.

---

### Decision · Program_Chairs · 2023-01-20

**Decision:**

Accept: poster

**Justification For Why Not Higher Score:**

The results are interesting, but the paper does not have outstanding experiments or theory worthy of showcasing as a spotlight/oral.

**Justification For Why Not Lower Score:**

The authors address important limitations on prior work on using VAEs to learn state-action space representations for formally verifying properties of deep RL policies.

**Metareview: Summary, Strengths And Weaknesses:**

The authors propose a novel discrete latent space VAE to represent the state-action space in a reinforcement learning problem. Doing so enables the authors to obtain a more efficient algorithm than prior work while also obtaining formal guarantees on their approach via the theory of bisimulation.

Strengths:
1. The authors systematically address shortcomings in prior work on using VAEs to represent state-action spaces.
2. They obtain more efficient algorithms with formal guarantees based on bisimulation.

Weaknesses:
1. Minor weaknesses around clarity/presentation were raised by reviewers in the initial version of the paper that were adequately addressed during the rebuttal phase.

Hence, I recommend acceptance.

**Note From Pc:**

if the above contains the word "oral" or "spotlight" please see: "oral" presentation means -> notable-top-5% and "spotlight" means -> notable-top-25%. As stated in our emails, we are disassociating presentation type from AC recommendations

**Summary Of Ac-Reviewer Meeting:**

No meeting.